

# A comprehensive evaluation of enhanced temperature influence on gas and aerosol chemistry in the lamp-enclosed oxidation flow reactor (OFR) system

Tianle Pan[1,2,3,4,5], Andrew T. Lambe[6], Weiwei Hu[1,2,4,5], Yicong He[7,a], Minghao Hu[8], Huaishan Zhou[1,2,3,4,5], Xinming Wang[1,2,4,5], Qingqing Hu[9], Hui Chen[9], Yue Zhao[10], Yuanlong Huang[11], Doug R. Worsnop[6,12], Zhe Peng[13,14], Melissa A. Morris[13,14], Douglas A. Day[13,14], Pedro Campuzano-Jost[13,14], Jose-Luis Jimenez[13,14], Shantanu H. Jathar[6]

[1]State Key Laboratory of Organic Geochemistry, Guangzhou Institute of Geochemistry, Chinese Academy of Sciences, Guangzhou 510640, China
[2]CAS Center for Excellence in Deep Earth Science, Guangzhou, 510640, China
[3]Chinese Academy of Sciences University, Beijing 100049, China
[4]Guangdong-Hong Kong-Macao, Joint Laboratory for Environmental Pollution and Control, Guangzhou Institute of Geochemistry, Chinese Academy of Science, Guangzhou 510640, China
[5]Guangdong Provincial Key Laboratory of Environmental Protection and Resources Utilization, Chinese Academy of Science, Guangzhou 510640, China
[6]Aerodyne Research Inc., Billerica, Massachusetts, 01821, United States
[7]Department of Mechanical Engineering, Colorado State University, Fort Collins, Colorado 80523, United States
[8]China-UK Low Carbon College, Shanghai Jiao Tong University, Shanghai, 201306, China
[9]Key Laboratory of Organic Compound Pollution Control Engineering, School of Environmental and Chemical Engineering, Shanghai University, 200444, Shanghai, China
[10]School of Environmental Science and Engineering, Shanghai Jiao Tong University, Shanghai, 200240, China
[11]Department of Environmental Science and Engineering, California Institute of Technology, Pasadena, California 91125, United States
[12]Institute for Atmospheric and Earth System Research (INAR) / Physics, Faculty of Science, University of Helsinki, Helsinki, 00014, Finland
[13]Cooperative Institute for Research in the Environmental Sciences (CIRES), University of Colorado at Boulder, Boulder, Colorado, 80309, United States
[14]Department of Chemistry, University of Colorado at Boulder, Boulder, Colorado 80309, United States
[a]now at: State Key Joint Laboratory of Environmental Simulation and Pollution Control, School of Environment, Tsinghua University, Beijing 100084, China

*Correspondence to*: Weiwei Hu (weiweihu@gig.ac.cn)

**Abstract.** Oxidation flow reactors (OFRs) have been widely used to investigate the formation of secondary organic aerosol (SOA). However, the UV lamps that are commonly used to initiate photochemistry in OFRs can lead to increases in the reactor temperature with consequences that have not been assessed in detail. In this study, we systematically investigated the temperature distribution inside an Aerodyne Potential Aerosol Mass OFR and the associated impacts on flow and chemistry arising from lamp heating. A lamp-induced temperature enhancement was observed, which was a function of lamp driving voltage, number of lamps, lamp types, OFR residence time, and positions inside OFR. Under common OFR operational conditions (e.g., < 5 days of equivalent atmospheric OH exposure under low-NOx conditions), the temperature enhancement was usually within 1-5 °**C**. Under extreme (but less commonly used) settings, the heating could reach 15 °**C**. The influence of the increased temperature over ambient conditions on the flow distribution, gas, and condensed phase chemistry inside OFR was evaluated. We found that the increase in temperature changes the flow field, leading to a reduced tail on the residence time distribution and corresponding oxidant exposure due to faster recirculation. According to simulation results from a box model using radical chemistry, the variation of absolute oxidant concentration inside of OFR due to temperature increase was small (<5%). The temperature influences on existing and newly formed OA were also investigated, suggesting that the increase in temperature can impact the yield, size, and oxidation levels of representative



biogenic and anthropogenic SOA types. Recommendations for temperature-dependent SOA yield corrections and OFR operating protocols that mitigate lamp-induced temperature enhancement and fluctuations are presented. We recommend blowing air around the outside of the reactor with fans during OFR experiments to minimize the temperature increase inside OFR. Temperature increases are substantially lower for OFRs using less powerful lamps than the Aerodyne version.

## 1 Introduction

Secondary organic aerosol (SOA) can account for 60-95% of OA and 10-75% of submicron particles (Jimenez et al., 2009), thus, strongly impacting air quality (Huang et al., 2014), climate (Myhre, 2013; Poschl, 2005) and human health (Nel, 2005; Feng et al., 2016; Nault et al., 2021). Elucidating the formation mechanism of SOA is crucial for clarifying its environmental impact (Ziemann and Atkinson, 2012; Hallquist et al., 2009; Klyta and Czaplicka, 2020). To investigate SOA formation, chambers (Hildebrandt et al., 2009; Cocker et al., 2001; Paulsen et al., 2005; Wang et al., 2014; Carter et al., 2005; Martin-Reviejo and Wirtz, 2005; Rollins et al., 2009; White et al., 2018; Zong et al., 2023) and flow tubes (Cooper and Abbatt, 1996; George et al., 2007; Hanson and Lovejoy, 1995; Robbins and Cadle, 2002; Katrib et al., 2005; Knopf et al., 2005; Ezell et al., 2010) have been commonly used in the laboratory or investigating secondary formation in the early periods of past several decades. In the last decade, due to the fast development of online measurement techniques, different types of oxidation flow reactors (OFR), which are portable and can be used in field studies to explore SOA formation under ambient conditions, have been developed and widely deployed (Kang et al., 2007; Watne et al., 2018; George et al., 2007; Peng and Jimenez, 2020; Lambe et al., 2011b; Shah et al., 2020; Saha et al., 2018; Xu and Collins, 2021; Chu et al., 2016; Simonen et al., 2017; Li et al., 2019; Keller et al., 2022 and references therein).

The potential aerosol mass (PAM) reactors, initially distributed by Prof. Bill Brune (the Penn State PAM) and later modified and commercialized by Aerodyne Research Inc. are the most widely used OFRs for studying SOA formation and evolution under ambient environments (Peng and Jimenez, 2020). The PAM reactor was first proposed and designed by Kang et al. (Kang et al., 2007) and has been successfully deployed in a variety of field studies including forests (Hu et al., 2016; Palm et al., 2016; Palm et al., 2017; Palm et al., 2018; Sumlin et al., 2021), urban areas (Ortega et al., 2016; Liu et al., 2018; Chen et al., 2021; Hu et al., 2022; Sbai et al., 2021; Park and Kim, 2023; Xu et al., 2022; Park et al., 2019), rural sites (Ahlberg et al., 2019; Hodshire et al., 2018; Hu et al., 2016), tunnel/roadside studies (Liu et al., 2019; Liao et al., 2021; Tkacik et al., 2014; Saha et al., 2018), as well as in multiple laboratory studies (Kang et al., 2007; Lambe et al., 2011b; Bahreini et al., 2012; Bruns et al., 2015; Sengupta et al., 2018; Kramer et al., 2019; Cheng et al., 2021; Lei et al., 2022; Srivastava et al., 2023 and references therein). In the PAM, high concentrations of the OH radicals, which is the major oxidant for SOA formation in ambient air, can be generated quickly by initiating the photochemistry of $O_3$ and $H_2O$ with UV lamps (185/254 nm) mounted inside of a flow tube (Lambe et al., 2011b; Li et al., 2015; Peng et al., 2015).

Earlier studies using the Penn State PAM reported a temperature increase of 2 °C. However, several recent studies using the PAM show that the UV lamps can increase the internal temperature of that reactor by 0-15 °C (up values are under extremely high voltage settings) above ambient temperature (Lambe et al., 2011a; Chen et al., 2013; Tkacik et al., 2014; Lambe et al., 2019; Charnawskas et al., 2017). However, a full investigation of the lamp heating effect on the temperature distribution inside of PAM-OFR and clarification of its influences on the flow distribution inside of PAM-OFR has not been reported yet. The temperature is a key parameter for gas diffusion and gas/aerosol partitioning (Pankow, 1994; Donahue et al., 2006). The increase in the temperature inside an OFR can result in the formation or deepening of recirculating flow, which leads to a shorter residence time and a broader residence time distribution (Huang et al., 2017; Lambe et al., 2019). Numerous chamber studies have found that higher temperatures may substantially decrease the SOA



yield (Gao et al., 2022; Lamkaddam et al., 2017; Boyd et al., 2017; Price et al., 2016; Clark et al., 2016; Tillmann et al., 2010; Pathak et al., 2007) and influence SOA chemical composition (Jensen et al., 2021; Gao et al., 2022; Simon et al., 2020; Li et al., 2020a; Kristensen et al., 2020; Quéléver et al., 2019; Denjean et al., 2015). Thus, the increase of temperature inside of PAM-OFR, which could lead to extra uncertainty for SOA simulation and study, should be further assessed and clarified.

In this study, we systemically investigate the effects of lamp-induced heating on PAM-OFR measurements. Based on computational fluid dynamics (CFD) simulation, we show how the temperature affects the flow and average OH exposure inside the PAM-OFR. Two box models were applied to illustrate how the enhanced temperature impacts the chemistry of gas-phase reactions and SOA formation inside PAM-OFR. The model results can be used as a rough reference for correcting the related experimental results due to enhanced temperature. Finally, we propose the use of fans to cool the

OFR and an online sampling strategy that alternates high and low driving voltages at varied lamp settings to reduce the uncertainty caused by the varied heating effect.

## 2 Methods

### 2.1 Oxidation flow reactor (OFR)

The PAM OFR (Aerodyne Research, Inc.) used in this study is a 13 L horizontal aluminum cylindrical chamber (46

cm long × 22 cm ID). Four low-pressure germicidal mercury (Hg) lamps are isolated from the sample flow by type 214 quartz tubes (Fig. S1). The Hg UV fluorescent lamps used in this study are commercial products (GPH436T5VH/4 or GPH436T5L/4, Light Sources, Inc.) which are the default light bulbs chosen by ARI for the OFR (Fig. S2). In addition, the temperature within an OFR with the Penn State low-pressure Hg UV lamps (model no. 82-9304-03, BHK Inc.), which are also widely used in PAM-OFR (Khalaj et al., 2021; Xu and Collins, 2021; Lambe et al., 2011b; Siemens et al., 2022;

Hu et al., 2016; Ortega et al., 2013; Link et al., 2016; Palm et al., 2016; Kang et al., 2018; Mitroo et al., 2018; Kang et al., 2007), are also measured. For the Light Sources lamps, the light intensity of each lamp was changed through the AC voltage input to the lamp ballast, controlled by a computer with settings ranging from 0 V (minimum) to 10 V (maximum, full AC output). For BHK lamps, the maximum setting voltage is 5 V (full AC output). At maximum voltage, the best estimation of supply power is 8.9 W (see sect. S1 for details) to each lamp for the Light Source lamp and 6.3 W for the BHK lamp.

When the lamps are turned on, a pure nitrogen purge gas flows through the lamp sleeve at a flow rate of 0.2-0.3 L min$^{-1}$. The lamp sleeve is defined as the space between the lamps and the wall of the quartz tube. This is done to avoid the accumulation of oxidation products on the surface of the lamps and to somewhat cool the UV lamps (minor effect, as discussed below). On the inner surface of the front plate of the OFR, a hexagon nut is connected to the center inlet. The side of the nut is drilled with holes to promote axial mixing of the sample flow, and is hereafter referred to as an inlet

diffuser (Fig. S1c) (Mitroo et al., 2018). Two mesh screens, used to block debris and insects and make the air flow more even (helping to break up eddies from outside), are installed inside of the front and back plates, respectively (Fig. S1b). A detailed schematic plot of the OFR used in this study can be found in Fig. 1.

### 2.2 Measurement of OFR temperature

The temperature distribution inside OFR was measured with multiple temperature sensors. The types and models of

temperature sensors used are listed in Table S1. To assess the accuracy of temperatures measured by different temperature sensors, a comparison experiment was carried out under different ambient environments. We found good agreement (within





2 °C) among different temperature sensors, supporting the robustness of the temperatures measured in this study (Fig. S3). Detailed information on experimental settings for temperature measurements inside the OFR is summarized in Fig. 1 and Table S2. Briefly, we measured the air temperature inside OFR at varied positions (vertically and horizontally) under

different lamp settings (e.g., number of lamps, lamp types, and lamp intensity) and flow rates. To achieve this, an external temperature sensor was placed directly inside the OFR and was moved manually from outside. The values reported in this study were obtained after the temperature inside OFR was stable for ~20 min after switching OFR settings. During the measurement, the inlet diffuser and the mesh screen in the front plate of OFR or in the backplate needed to be removed to extend the thermocouple inside the OFR. We have tested if the removal of the mesh screen would change the temperature

field inside OFR and found a negligible impact (Fig. S4 (removed the back plate mesh screen) vs. Fig. 5 (removed the inlet diffuser and front plate mesh screen)). We also covered the sensor with shielding paper to avoid the potential influences of optical radiation from the UV lamps on temperature detection during this process. In addition to the inside temperature of OFR, the temperature of the inner/external surface of OFR, and the temperature in the gas output of OFR were also measured for comparison. A sensor for detecting temperature and relative humidity (RH) (referred to as a "primary T/RH

sensor", Sensirion SHT21, Sensirion AG) was mounted on the back plate at the factory. Thus, the default primary T/RH sensor was used for temperature measurement for all the experiments.

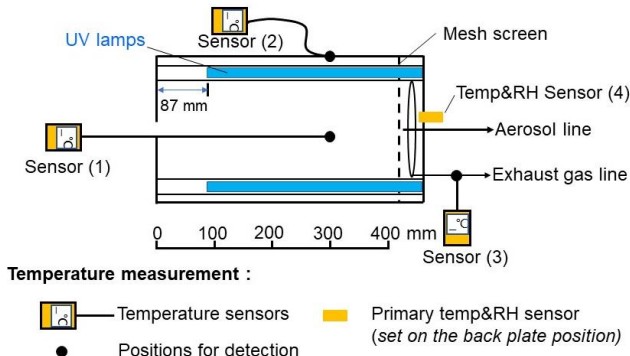

**Figure 1: Schematic plot for temperature measurement in the oxidation flow reactor of this study. The center inlet,**

**nut, and mesh screen near the front plate were removed when the temperature sensor was probed in the front direction. The information of different temperature sensors used can be found in Table S1.**

### 2.3 Model simulations on temperature distribution and flow field inside OFR

Computational Fluid Dynamics (CFD) simulations were performed using the ANSYS Fluent software (Version 14.5) in three dimensions to simulate the temperature distribution and flow field in OFR. ANSYS has been used to simulate the

flow field inside flow reactors in past studies (Li et al., 2019; Ihalainen et al., 2019). The temperature distribution and flow field under unheated and heated conditions were both simulated with the CFD model. The simulated experimental condition is 5 L min$^{-1}$ of air as the carrier gas and 0.3 L min$^{-1}$ of $SO_2$ which was injected for 2 s as the tracer gas. The convergence of this model was defined when the residuals of physical quantities (e.g., pressure, temperature, velocity, density, and viscosity, etc.) were below 0.001.

For the unheated condition, the more advanced "realizable k-epsilon turbulence model" was used (Shih et al., 1995). The simulation was solved by using the pressure-based SIMPLEC algorithm, which is a commonly used method in CFD



models for simulating incompressible flow problems (Patankar and Spalding, 1972), as OFR conditions with lights were off. For the heated condition, the thermal boundary was set to a series of fixed temperatures, based on those directly measured in the OFR. When the driving voltage of the OFR lamps was all set to be 5 V, the model settings applied were

55 °C for the surface of the quartz UV lamp sleeves, 35 °C for the inner wall surface, and 26 °C for the carrier gas. When all the voltages of the OFR lamps were 10 V, those values were set to 62 °C, 40.5 °C, and 23 °C, respectively. The lower carrier gas temperature under 10 V conditions vs, 5V was caused by the different room temperatures in each real experiment. The viscosity and thermal conductivity of mixing gas were calculated using the mass-weighted-mixing-law (Ni et al., 2010). Least Squares Cell-Based method and species transport model were used for the solution (Ghia et al., 1982).

**2.4 KinSim kinetic model for gas-phase reactions**

The influence of temperature on gas-phase reactions was modeled with a box model (KinSim 4.14 in Igor Pro. 6.37) with the OFR radical mechanisms used in Li et al. (2015), Peng et al. (2015) and Peng et al. (2019). All the gas-phase reactions were obtained from the JPL chemical kinetic data evaluation, which includes temperature as a variable for reaction rate calculation (Burkholder et al., 2019). In the model, we assumed a mixing volume of 800 ppb $SO_2$ injection into OFR

under a constant 2.2% water vapor mixing ratio (RH=70% at 25 °C) based on calibration experiment settings. This corresponds to an OH reactivity (OHR) of the incoming air of 20 $s^{-1}$. The simulated $SO_2$ output concentration in the model was weighted by the measured residence time distribution (RTD) at a flow rate of 5 L $min^{-1}$ (~700 s). Good agreement between modeled $SO_2$ decay based on the KinSim model and measured $SO_2$ results has been shown in Hu et al. (2022).

The mechanism of OFR185 mode, in which $O_3$ and OH radicals were generated by the photolysis of $O_2$ and $H_2O$ inside

OFR, was applied in the model. Note that the ratio of photon flux between 185 and 254 nm from lamps produced by Light Source Inc. does not change as a function of their intensity (Rowe et al., 2020), while these ratios do vary with intensity for the BHK lamps (Li et al., 2015). In the model, the photon flux ratio of 185/254 nm was set to be constant (5%). Results for 25 °C to 40 °C (binned with 5 °C) were modeled, which covers the typical temperature range inside OFR under most lamp setting cases. These simulations were done with RTD obtained under unheated conditions in OFR (25 °C). Since the

residence time distribution (RTD) inside OFR can also be influenced by temperature (Lambe et al., 2019), we also show the model results with measured RTD at 40 °C.

**2.5 SOM model for SOA formation**

The statistical oxidation model (SOM) is a kinetic, process-level model to simulate SOA formation by accounting for gas-particle partitioning, multi-generational oxidation (fragmentation and functionalization), and autoxidation for highly

oxygenated molecules (HOMs) (Cappa and Wilson, 2012; Eluri et al., 2018; He et al., 2021; He et al., 2022). The SOM model has been shown to successfully capture the evolution of SOA formation and oxidation in multiple laboratory and field studies (Cappa and Wilson, 2012; Jathar et al., 2015; Eluri et al., 2018; Akherati et al., 2020; He et al., 2021; He et al., 2022). In the SOM model, the oxidation of a VOC precursor is tracked by its evolution within a carbon−oxygen grid. The volatilities and reactivity ($k_{OH}$) of organic species in each grid are calculated based on their carbon ($N_C$) and oxygen

numbers ($N_O$) (Cappa and Wilson, 2012; Jathar et al., 2015; Eluri et al., 2018). In total, six adjustable parameters are used to determine the reaction probabilities and volatility of the grid species. Four parameters (p1-p4, Table S3) indicate the number of oxygens added per functionalization reaction, and two other parameters describe the probability of fragmentation ($m_{frag}$, Table S3) and the decrease in vapor pressure ($\Delta LVP$, Table S3) per oxygen atom added to the carbon backbone (Jathar et al., 2016; Akherati et al., 2019).





In this study, we used the SOM model coupled with the TwO-Moment Aerosol Sectional model (TOMAS) (Adams and Seinfeld, 2002; Pierce et al., 2007) to simulate the temperature influences on the yield, oxidation state, and size distributions of newly formed SOA in the OFR. In the model, the temperature dependence is mainly calculated based on the Clausius–Clapeyron Equation:

$$P_i^* = P_{i,ref}^* \times e^{\left(\frac{H_i^{vap}}{R} \times 10^3 \times \left(\frac{1}{298} - \frac{1}{T}\right)\right)} \tag{1}$$

where $R$ (8.314 J mol⁻¹ K⁻¹) is the ideal gas constant, $P_i^*$ and $P_{i,ref}^*$ (Pa) are the saturation pressure of species $i$ at target temperature and reference temperature (298 K used here), respectively. The $P_{i,ref}^*$ can be calculated based on the following equation (2):

$$P_{i,ref}^* = \frac{C_{i,ref}^* \times R \times 298}{MW_{org,i} \times 10^6} \tag{2}$$

where $MW_{org,i}$ (g mole⁻¹) is the molecular weight of organic species $i$ and $C_{i,ref}^*$ (μg m⁻³) is the saturation
concentration at 298 K. The $MW_{org,i}$ and $C_{i,ref}^*$ are calculated based on the number of C ($N_C$) and O ($N_O$) and $\Delta LVP$ of species $i$ (Eluri et al., 2018). The $H_i^{vap}$ (kJ mol⁻¹) is the evaporation enthalpy of species $i$ calculated as followed equation (Epstein et al., 2010):

$$H_i^{vap} = -11 \times log\, C_{i,ref}^* + 131 \tag{3}$$

In addition, constant $H_i^{vap}$ of 80, 100, 120 and 150 kJ mol⁻¹, derived from analysis of field data, was also tested in
the model (Saha et al., 2017; Cappa and Jimenez, 2010; Louvaris et al., 2017).

In this study, SOA formation from four typical VOC precursors including dodecane, α-pinene, toluene, and m-xylene under different OA concentrations (1-80 μg m⁻³) and NOx conditions (low NOx vs high NOx) was modeled. Five stages of temperature ranging from 20-40 °C were simulated. Similar to the gas-phase simulations, results were also calculated based on the measured RTD obtained at both ~25 °C and ~40 °C (for the modeled yields under 40 °C) with a total model
time of 700 s. Additionally, we also considered the highly oxygenated organic molecules (HOMs) in this model that contribute to new particle formation (He et al., 2021; Bianchi et al., 2019). The HOM yields and other detailed information for input parameters can be found in Table S3.

### 3 Results and discussions

### 3.1 Heating effect inside OFR

To characterize the temperature distribution in OFR, we measured the temperature as a function of lamp driving voltage, number of lamps used, flow rate, lamp types, and different space positions.

### 3.1.1 Enhanced temperature vs lamp driving voltage (OH exposures) and flow rate

In general, we observed a systematic increase in temperature inside the OFR with Light Source lamps. As shown in Fig. 2a, the enhanced temperature in the OFR increases as a function of lamp driving voltage and, hence, OH exposures.
When the driving voltage on the 2-lamp setup was changed in increasing fashion (from 0 V to 3 V, Light Source lamp), the temperature enhancement in the OFR was generally lower than 5 °C. 3 V is roughly equivalent to a photochemical aging time of around 5 days (assuming the water mixing ratio is 1.88%, RH≈60%, external OH reactivity=30 s⁻¹). For a





higher voltage range (from 3 V to 10 V, which is equivalent to a photochemical aging time from 5 days to two weeks), the temperature inside the OFR increased by 10 °C. Note that the relationship between the lamp driving voltage and OH

exposure varied with the number and type of UV lamps used, as well as how the OFR was operated, as shown in Fig. S5. When four lamps were turned on and the driving lamp voltage was increased from 2 to 10 V, the temperature increased by 5-20 °C inside the OFR for the laboratory tests (Fig. 2a and 3a). These results support that the temperature increase inside of the OFR is mainly due to the heat from the lamps and was further supported by Fig. 3b, which shows that the temperature enhancement inside the OFR decreased as a function of the flow rate from 3 L min⁻¹ to 7 L min⁻¹ (Fig. 3b). The anti-

correlation between the temperature enhancement and the flow rate was mainly due to the higher heat capacity provided by the increased air flow, thus needing a lower temperature increase to carry the excess heat from the lamps.

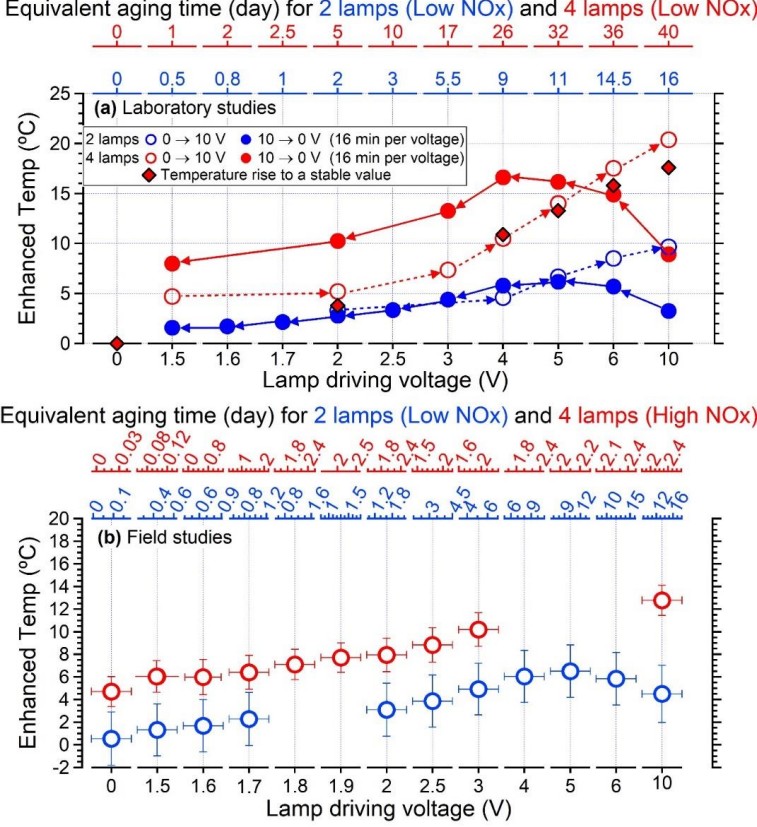

**Figure 2: (a) Enhanced temperature inside OFR (Measured temperature in OFR minus ambient temperature) as a function of the lamp driving voltage using two lamps and four lamps in the laboratory studies. Note that without**

**specific notification, all the temperature measurement results shown in the main text are obtained with lamps produced by Light Source Inc. The relationship between lamp driving voltage and OH exposures for using two lamps and four lamps can be found in Fig. S5. The OH exposures were calculated based on the empirical parameters in Hu et al. (2022) assuming a mixing water ratio of around 1.88% and external OH reactivity of 30 s⁻¹. The equivalent aging days (top axis) were estimated by assuming ambient OH concentrations are around 1.5x10⁶**



**molecules cm⁻³. The lamp type applied here is uncovered for field studies with four lamps in Fig. 2b and 80% covered for other conditions. The measured position of temperature is in the centerline with 300 nm probing. The flow rate is 5 L min⁻¹. (b) Enhanced temperature inside OFR as a function of light settings in the field studies. Two lamps were applied in the low NOx conditions, while the four lamps were for high NOx conditions. The high NOx reaction regimes were achieved by inputting extra N₂O as illustrated in Peng et al. (2018) and Lambe et al. (2017).**

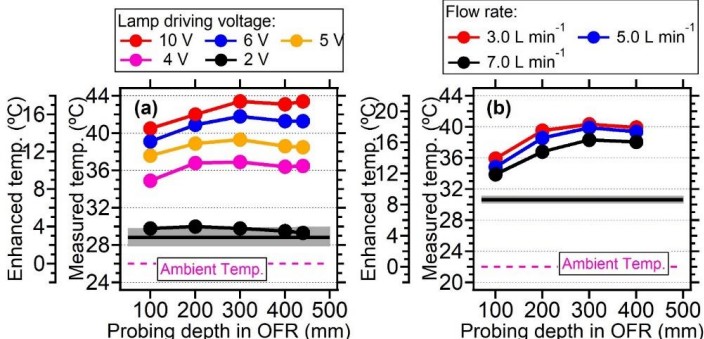

**Figure 3: (a) The measured temperature under different lamp driving voltage and (b) at different flow rates for Light Source lamp. The temperature was measured in the centerline at different probing positions. Black lines and gray shadows represent the temperature measured by the primary T/RH sensor. For panel (a) four lamps were all turned on during measurement. The flow rate is 5 L min⁻¹; For panel (b), four lamps were set in 10 V.**

When the lamp driving voltage was set in a monotonically decreasing fashion (e.g., from 10 V to 0 V), the maximum temperature increase did not occur at the highest voltage setting of lamps (10 V), but in the range of 4-6 V, as shown in Fig. 2a and b. This inconsistency is mainly due to the lamps starting at 10 V with colder conditions (e.g., room temperatures or lower voltage settings), meanwhile, the OFR reactor has a thermal mass that needs time to accumulate or dissipate heat. Fig. 4 shows that the entire system needs more than 100 minutes to achieve a thermal steady state. However, a much shorter

time (15-24 minutes) is usually applied in the laboratory and field studies when voltages are systematically varied (Link et al., 2016; Murschell and Farmer, 2018; Hu et al., 2016; Ortega et al., 2016; Palm et al., 2016; Palm et al., 2018; Saha et al., 2018; Shah et al., 2020; Hu et al., 2022). These short constant lamp power times result in reduced temperature variation between different power settings, as well as inconsistency between maximum temperature peak with maximum power setting. To be clear, the time series of measured temperature when the lamp driving voltage was set under a representative

lamp voltage cycle (decreasing fashion: 10-6-5-4-2-0 V) are displayed in Fig. 5. Two scenarios with two and four lamps being turned on were both shown. The reason that the lamp voltage starts with 10 V in this typical setting is to maximize the efficiency of filament ignition in UV fluorescent lamps. Even though the thermal steady state was not achieved at each voltage setting, temperature increases of 2-8 °C for two lamps and 6-16 °C for four lamps were observed. In general, the temperature enhancement inside OFR is a function of the voltage profile and the time spent at each voltage. Maximum

SOA formation is typically observed after 1-2 days of equivalent age (Ortega et al., 2016; Palm et al., 2016; Hu et al., 2022). In most cases, the temperature enhancement inside OFR is generally less than 5 °C below 5-7 days of equivalent atmospheric OH exposure under laboratory and field studies as shown in Fig. 2a and 2b. The main exceptions involve specific OFR modes that incorporate N₂O or isopropyl nitrite precursors to establish high-NOx conditions (Lambe et al., 2017; Lambe et al., 2019). In these cases, four lamps are beneficial to compensate for the absorption of 185 nm radiation



by N₂O (Peng et al., 2018) that hinders HO$_x$ production, or the relatively weak absorption of UVA radiation by isopropyl
nitrite, respectively. This also makes the radiation within the OFR more uniform, as when only two lamps are used, the
area with another two lamps off would be darker due to the absorption of radiation.

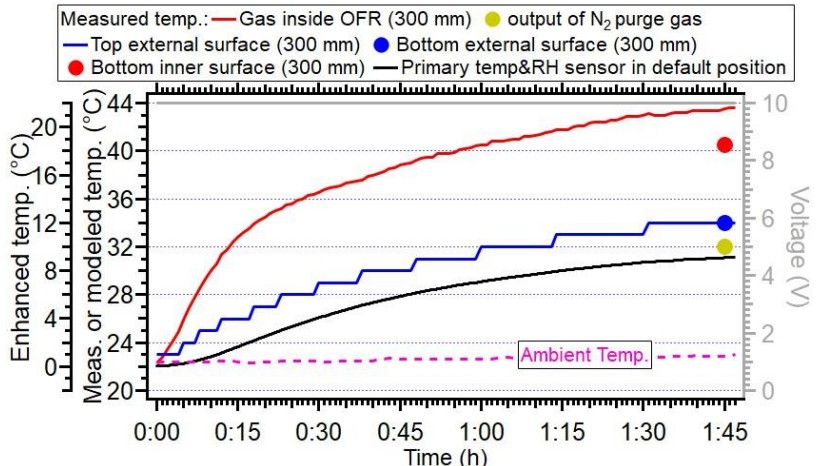

**Figure 4: The temperature variation under 10 V driving voltage for 4 lamps as a function of time. The start time is**
**the lamps being turned on. The temperature sensors were set at a depth of 300 mm in the central line to measure**
**the temperature inside OFR. The flow rate is 5 L min⁻¹. The temperature of the external surface of OFR is also**
**measured at the top and bottom positions. The temperature measured by the default OFR sensor installed in the**
**backplate is also shown in the black line. The purple dotted lines represent the ambient temperature.**

**3.1.2 Enhanced temperature vs lamp types.**

The temperature enhancement as a function of different types of lamps was also investigated here. The four types of
tested lamps are shown in Fig. S2. Fig. S6 shows similar temperature distribution inside OFR no matter whether 80%
covered or uncovered lamps, as well as no matter whether only 254 nm or 254/185 nm lamps were used. These results
suggest that the heating inside OFR is mainly from the heat transfer of the hot quartz sleeve (heated by the lamps) but not
from their optical radiation. Thus, different types of lamps impact mostly the OH exposures, but not the heating effect. The
three types of lamps applied here are all from the same manufacturer (Light Source Inc.) and, therefore have a similar value
of power supplied to each lamp (8.9 W). In addition, we also tested the temperature enhancement in the OFR with lamps
from another producer (BHK Inc.). An enhanced temperature of 6 °C inside the OFR was found at the centerline when two
BHK lamps at full power (Fig. S7b) were used in an OFR with a flow rate of 4.5 L min⁻¹. The less enhanced temperature
with BHK lamps is probably due to the lower input heating energy (6.3 W per lamp) than the lamp from Light Sources.
For the lower driving voltage of 0.95 V for BHK lamps as run during a field campaign, a temperature enhancement of 1-
2 °C with an equivalent aging time of around 1.5 days was observed (Fig. S7a and S7b). For that campaign, the OFR was
operated mostly continuously at ~1.5 days equivalent aging, since most SOA formation often is observed at these moderate
exposures, and is low enough that heterogeneous oxidation is not yet significant (e.g., Palm et al. (2016), Ortega et al.
(2016) and Hu et al. (2016)).

**3.1.3 Mapping out the temperature enhancement inside the OFR**



In this section, the enhanced temperatures inside of the OFR as a function of position are explored. In our case, the voltages of four lamps are set to be 5 V (~30 days' equiv. aging time and flow rate of 5 L min$^{-1}$) the temperature in the OFR is generally 9-15 °C higher (Fig. 6a) than ambient (26 °C). In the vertical axial direction, the enhanced temperature is higher at the top position (~14 °C) than at the bottom (~9 °C) due to the warm air ascending in OFR resulting from its

lighter density. We also found a similar conclusion for BHK lamps (Fig. S7) with lower temperature enhancement and smaller temperature difference in the vertical direction for the same equivalent aging. Horizontally, the temperature distribution is symmetrical, which shows a slightly lower temperature value in the middle and enhanced temperature on the edges (Fig. 6b). These temperature values were measured until their reading varied no more than 0.1 °C. The lower center temperature is probably caused by the faster flow rate (shorter residence time) of air mass in the middle due to the

removal of the inlet diffuser and the longer distance from the lamps. However, the largest temperature gradient at different horizontal axial positions is within 2-3 °C, suggesting the general temperature distribution measured here is similar to the distributions when the inlet diffuser is installed. For different probing depths, Fig. 3 generally shows the enhanced temperature is lower near the inlet and higher from the middle position to the back in the central line, which is mainly due to that the airflow flowing from the inlet to the outlet is being warmed, as well as the set position of lamps has a gap with

the front plate with the input inlet, as demonstrated in Fig. 1.

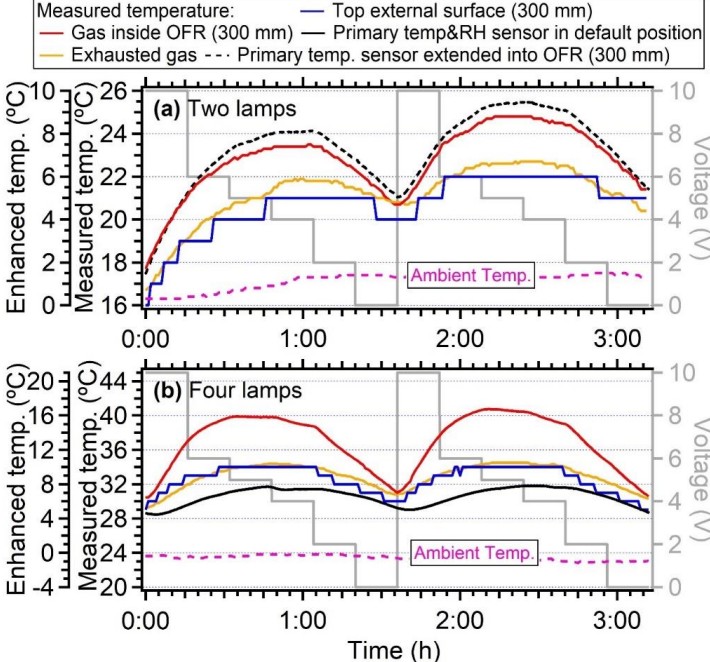

**Figure 5: Temperature variation under different lamp driving voltage cycled as 10-6-5-4-2-0 V for: (a) Two symmetrical lamps on the diagonal direction were used and the OFR backplate sensor was extended into OFR (from the backplate) at a depth of 300 mm from the inlet. (b) Four lamps were used and the OFR sensor was set in the**

**backplate. Each lamp setting lasts for 16 min. The flow rate of sampling air is 5 L min$^{-1}$ (residence time is 167 s).**

**3.1.4 Loss of heating energy in OFR**



The gain of the heating energy inside the OFR generally comes from the UV lamps, while the energy loss inside the OFR is mainly through three pathways: (1) The loss of energy through exhaust air of OFR; (2) The dissipation of heat energy from the OFR chambers (OFR metal tubes) to the ambient air by convection and radiation. (3) The energy loss
through purge nitrogen between the lamps and quartz sleeves (Fig. S8).

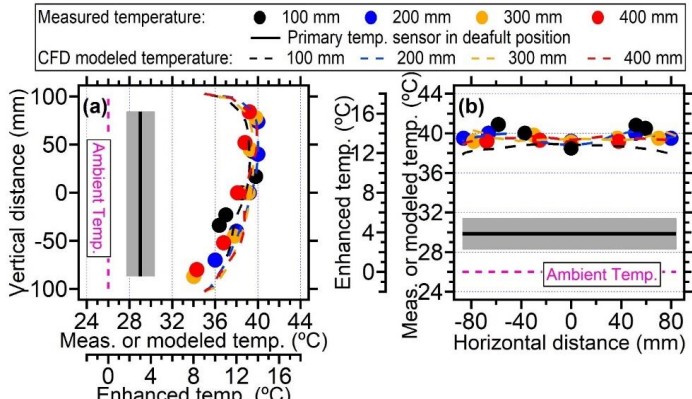

**Figure 6: The temperature measured by external temperature sensor at different positions inside OFR or under different settings: (a) the measured and CFD modeled temperatures on the vertical direction at different depths; (b) the measured and CFD modeled temperatures on the horizontal direction at different depths. All the four lamps**
**were turned on during measurement. The four lamps' driving voltages were 5 V. The black lines and grey shadings are the average ± standard deviation for temperature measured by the backplate sensor.**

To clarify the energy balance of the three pathways, we measured the temperatures of the OFR tube, the purge nitrogen, and the exhaust air. In this case, we demonstrate the results under the maximum lamp driving voltage of four UV lamps (10 V) and at a flow rate of 5 L min$^{-1}$. Figure 4 displays the measured temperature of the OFR tube as a function of
time and at different positions (inner vs external surface, bottom vs top surface). The temperature of the top external surfaces increases as the gas temperature inside OFR gets higher, supporting that the OFR tube receives the heating energy from lamps. The temperature detected at the inner surface of the OFR (40.5 °**C**) is lower than the temperature of gases (43.5 °**C**), however, higher than the outside of tube surfaces (34 °**C**). The lower temperature of the external surface of the OFR than the air masses inside of the OFR is probably because i) the aluminum OFR chamber has a higher thermal mass than the air. Although the specific heat capacity of metal (0.908 J g$^{-1}$ K$^{-1}$ at 301.60 K) is similar to that of air (1.005 J g$^{-1}$
K$^{-1}$ at 300 K) (Giauque and Meads, 2002; Kieffer, 1956), the flow tube is much heavier than the air due to much higher density of the former (2700 Kg m$^{-3}$ vs 1.29 Kg m$^{-3}$), resulting in a lower temperature of OFR tube than the inner air; ii) there are heating losses on the surface of OFR tube since the ambient air temperature (23 °**C**) is much lower than OFR tubes resulting in heat transfer. This heat transfer is also reflected by the decreasing temperature gradient between inner
and external OFR tube surfaces (40.5→34 °**C**). We also measured the temperature of purged nitrogen at a default flow rate of 0.2 L min$^{-1}$. In this case, the temperature of exhausted nitrogen is ~32 °**C** (output), which is 9 °**C** higher than the temperature of input nitrogen (23 °**C**).

In addition to the air inside OFR, the temperature of exhausted air from OFR was also examined closely. A temperature sensor was set in the Teflon Tee connector near the output of the OFR backplate. We found a slightly lower temperature
enhancement (≤5 °**C**, Fig. 5a) in the exhausted air than the values of air measured directly inside OFR, which is probably



influenced by the colder surfaces of the fitting or sampling lines. Using the measured temperatures shown above, we can roughly calculate the fraction of energy loss among three pathways. Detailed information on the calculation process can be found in Sect. S2 of the supporting information. Those model results suggest that after 105 min of heating up, 51% of the total power (35.6 W, 8.9 W for each lamp, 4 lamps in total) during the heating process is converted into energy that leads

to the temperature increase of OFR. The remaining 49% of power is lost due to the conversion efficiency or converted into other energy that does not cause the warming of airflow inside OFR. For the fraction (51%) of energy that causes warming, the fraction of energy loss through purged nitrogen gas (at flow rate 0.2 L min$^{-1}$), heated air, and OFR tubes is 0.3%, 9.8%, and 89.9%, respectively, as shown in Fig. S9a. This suggests the heating transfer through metal tubes would be the dominant pathway. Cooling the OFR tubes is a good way to keep the OFR near ambient temperature. As shown in the following sect.

3.5, with two fans blowing the OFR tube, the heat dissipation of the OFR tube increases, resulting in a much lower increase in temperature inside the OFR.

In some OFR systems, a higher flow rate of nitrogen-purged air was applied (Zhao et al., 2021; Bruns et al., 2015; Li et al., 2019). E.g., Li et al. (2019) set the purged nitrogen flow around 30 L min$^{-1}$ for their custom-designed OFR system with exterior lamps to keep the temperature around 25 °C. We tried to introduce a nitrogen gas flow at 20 L min$^{-1}$ through

the lamps to increase the energy loss for the ARI OFR. This is almost the maximum flow rate of purged nitrogen we can test due to the small inner diameter of the fitting connected to the lamp tubes. After nitrogen was injected at such a high flow rate, we found the temperature of the lamp sleeve, the inner surface of flow tubes, and gas inside OFR dropped from 62, 40.5, and 44 °C (four lamps at 10 V) to 42, 34 and 36.5 °C (36-51% decreases in temperature increase, with ambient temperature at 23 °C), respectively. Based on the measured temperature, we found the energy loss fraction through purged

nitrogen increased from 0.3% to 32% (Fig. S9b). This is helpful, but it still cannot balance the heating energy input from the UV lamps. The temperatures of air in the OFR (36.5°C) are still much higher than the room temperature (23 °C). The large consumption of pure nitrogen gas for running such a high flow rate would also be a challenge for long-term experiments, especially for field studies.

### 3.1.5 Artificially low temperature measured by primary T/RH sensor in ARI OFR

During the experiment, we found the primary T/RH sensor installed in the OFR backplate (see the sensor in Fig. S1) always showed a much lower temperature (by 1-14 °C) than the temperature sensor probing inside of OFR, as shown in Figs. 3-6. We suspect that the lower temperatures detected by the primary sensor at the default backplate position were mainly caused by the direct contact of this sensor with the metal backplate which is at lower temperatures. When the primary sensor was placed around 300 mm into the OFR (closer to the lamps than the centerline), a similar temperature

was measured by the primary T/RH sensor and the one probing into the OFR (maximum difference of 1 °C), as shown in Fig. 5a. This suggests the primary T/RH sensor in the backplate with default OFR settings might lead to underestimation of the temperature inside of OFR, which should be verified and corrected by the users based on the configuration of their instrument.

### 3.2 Temperature influence on the flow field

In this section, we discuss the impact of enhanced temperature on fluid dynamics inside OFR based on Computational Fluid Dynamics (CFD) simulations. Detailed setting parameters were introduced in Sect. 2.3. The temperature field where the voltages of four UV lamps were all set to 5 V was tested here. As shown in Figs. 6a and b, the simulated temperature distribution generally shows good agreement with the values measured directly with the temperature sensor inside OFR,



validating the reasonableness and reliability of both simulated and measured temperature distribution in OFR. In Fig. 7a,
the simulated 2-D temperature distributions display the hottest air (up to 45 °C) wrapped around UV lamps. The rest of the
air parcels have a very large vertical axial temperature gradient (~10 °C) and the air with higher temperatures between 35-
45 °C remains in the upper position of flow tubes. Such a high-temperature gradient will lead to substantial recirculation
inside OFR, based on the Richardson number calculation (Ri = 3974, which is far above 10 and indicates the existence of
turbulence) (Huang et al., 2017; Holman, 2010). Details of the Ri calculation can be found in Sect. S3 of the supporting
information.

      To investigate the non-isothermal effect on fluid dynamics inside the OFR, the simulated flow distributions with and
without heating effects are demonstrated in Figs. 7c-j. As illustrated in Figs. 7c-d, the velocity of flow is highest after
injection through an inlet diffuser, resulting in recirculation near the edge of the walls. The other place showing a higher
velocity of air is around the output of exit (Figs. 7d and S10), which is generally used for particle output measurement. In
addition to aerosol sampling, the air for gas phase measurements was usually sampled through a perforated ring flow
manifold in the back of OFR to obtain more even and better-mixed air (Fig. S1). Under the typical operating condition, the
flow distribution when both gas and aerosol were sampled might be different from that when only the aerosol line was
applied. However, adding the extra output of the gas phase would greatly add to the complexity of this CFD simulation.
Thus, to simplify the simulation, we set the airflow to be only sampled through the exit port located in the center of the
OFR backplate. Moreover, when the flow ratio between gas and aerosol lines is different, the flow distribution inside OFR
would be changed.

      When the UV light was on, the flow distribution changed substantially and had more flow bifurcation and recirculation,
as shown in Figs. 7e and 7f. There is a large recirculation in the lower part of the flow tube. A clearer demonstration of the
recirculation can be seen in Figs. 7g-j and Fig. S10. When no heat effect was introduced in the OFR, the 1-D flow profile
generally exhibited symmetry in the vertical direction (Fig. 7h). A slight distortion on the 1-D flow profile in the horizontal
direction is reasonable due to the influence of the random motion and pressure gradient of gas (Fig. 7g). When a vertical
axial temperature gradient exists, 1-D flow profiles are skewed due to the buoyancy of the warm air. The recirculation
effect manifested when the temperature gradient is larger, as well as when the flow rate is lower, thus can substantially
change the residence time distribution (RTD) and its average value ($\tau_{avg}$). $\tau_{avg}$ can be calculated as the sum of the integrals
of signal (such as the signal of $SO_2$ after introducing $SO_2$ pulses) over time versus the sum of the signals (Huang et al.,
2017).

      The average RTD values inside the OFR under heat and non-heat scenarios were also measured and simulated in Fig.
8. A 2 s pulse of 50 ppm $SO_2$ was injected into an OFR carrier gas of 5 L min⁻¹ and RH <10% in both the laboratory
experiment and simulation model. For laboratory measurement, $N_2$ was used as carrier gas to prevent the reaction between
$SO_2$ and the generated oxidant when UV lamps were turned on. Similar to the modeled results in Huang et al. (2017) and
the measured results in Lambe et al. (2019), the measured RTDs under higher temperatures in this study exhibit shorter
$\tau_{avg}$ (135-145 s at 5-10 V vs 177 s at 0 V) due to the acceleration of air upon heating. The acceleration is mainly reflected
in the early arrival time and shorter tails, as shown in Fig. 8. Huang et al. (2017) found that when the flow tube is under
non-isothermal conditions even a small temperature deviation (0.2 °C) can create secondary flow thus affecting the RTD
and $\tau_{avg}$ substantially. The variation of the RTD when lamps were turned on and off was also simulated in the CFD model,
as shown in Fig. 8. Compared with the measured results, the simulated RTD generally shows an early arrival time and
broader distribution at different light settings. The simulated $\tau_{avg}$ upon heating (173-180 s) is higher than the value when
lights were off (167 s), indicating the recirculation in the model introduced by heating prolongs the $\tau_{avg}$ rather than

reducing the $\tau_{avg}$. The higher $\tau_{avg}$ when lamps were on than off contrasts with the trend found in the measured $\tau_{avg}$

(Huang et al., 2017; Lambe et al., 2019). This inconsistency is probably because i) we only considered the airflow sampled from the center outlet in the backplate, but not the ring flow manifold, which caused more recirculation and ii) the recirculation was weighted more in the calculation than the real conditions, which also emphasizes the challenge of simulating the orthogonal forces caused by the forced convection by the pressure gradient (horizontal) and buoyancy-induced free convection (vertical) (Huang et al., 2017). Note the simulated $\tau_{avg}$ under lamp setting of 10 V (173 s) is lower

than the $\tau_{avg}$ under 5 V (180 s), the trend of which is consistent with the measured results, suggesting the enhanced gas diffusion upon heating was properly considered except the recirculation.

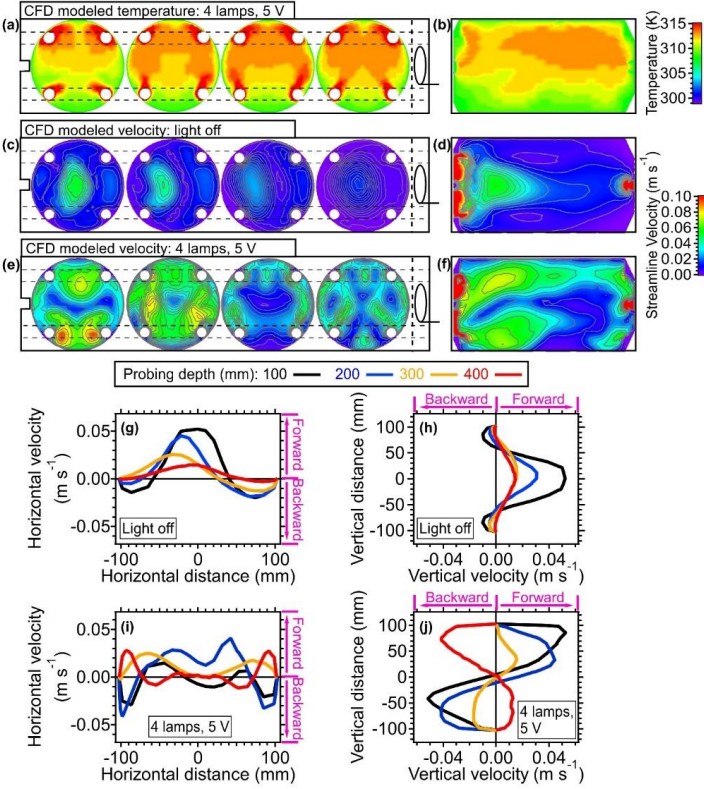

**Figure 7: (a) Three-dimensional simulation results demonstrating the cross-sectional temperature profiles. (b) Simulated lateral temperature profile inside of the OFR based on CFD simulation. Panel (a) and (b) were simulated**

**with conditions that four lamps were set to be 5 V. (c) Three-dimensional simulation results illustrating cross-sectional velocity profiles and (d) lateral velocity profiles based on CFD simulation results. Panel (c) and (d) were simulated with conditions that lamps are off (room temperature: 25 °C). (e-f) the same plots as panels (c) and (d) with four lamps being set on 5 volts. Enhanced temperature influences in panels (c) and (d) were considered. One-dimensional velocity profiles at (g) horizontal and (h) vertical directions inside the OFR at room temperature, One-**

**dimensional velocity profiles at (i) horizontal and (h) vertical directions under four lamps being set 5 volts.**

In summary, the heating introduced from four lamps set to 5 V can decrease $\tau_{avg}$ by 18-23% relative to dark OFR


experimental conditions. This is considered as an upper limit variation on $\tau_{avg}$ for a typical OFR setting where the voltage of lamps was usually below 3 V to obtain OH exposures less than 5 days. The variation of RTD changes the exposure time of gas/aerosol species in the OFR, which can impact the gas and particle oxidation conditions. In the following sections,

this impact will be systematically discussed.

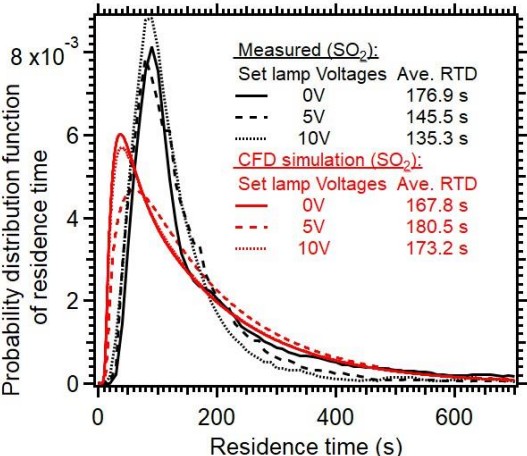

**Figure 8: Residence time distribution (RTD) of SO₂ inside OFR under different lamp settings. 2 s pulser of SO₂ was injected into the OFR. The average RTD values were also estimated here. The simulated results from CFD models are shown in red lines.**

**3.3 Temperature influence on gas-phase reaction and OH exposure**

In this section, the temperature influence on gas-phase reactions was systematically investigated. We take the oxidation of $SO_2$ inside OFR as an example to simulate its gas-phase reaction with oxidants (mainly OH) under a temperature range from 25 °C to 40 °C (binned with 5 °C) using the KinSim Model. The 25 °C simulation represents the typical laboratory condition, while 40 °C was the approximate temperature observed at the upper limit of lamp voltage (5-

10 V, corresponds to photochemical age ~10-30 days, Fig. 2). In addition, to account for the RTD influences on gas-phase reaction under different temperatures, the scenarios with measured RTD distributions under 25 °C and 40 °C were both simulated here.

In general, when the temperature variation inside of OFR (the same RTD as measured at 25 °C being applied for cases at different temperatures) was considered, the influences of temperature enhancement due to lamp heating on the gas-phase

reaction rate are very minor. The concentration of OH, $HO_2$, $O_3$, and $O(^1D)$ all showed a maximum of ~5% increase at the highest photon flux, which is consistent with the simulated results using the same KinSim model in Li et al. (2015). The $SO_2$ decay and OH exposure also show negligible variations in the model (Fig. 9e and 9f). When the temperature influences on the RTD were taken into account, the variation trend of oxidant concentration was mixed due to the combined negative effect of the reduced average residence time ($\tau_{avg}$) and the positive effect of temperature on oxidant concentration (Li et

al., 2015). The parameter that was most influenced is OH exposure, which shows 18-20% lower values at 40 °C than at 25 °C due to the shorter residence time ($\tau_{avg}$) upon heating. Our result suggests the increase of temperature inside of OFR





due to lamp heating would have a very minor to negligible impact on the gas-phase reactions, whereas parameters related to the RTD distribution (e.g., OH exposure) shall be considered in the current and future experiments.

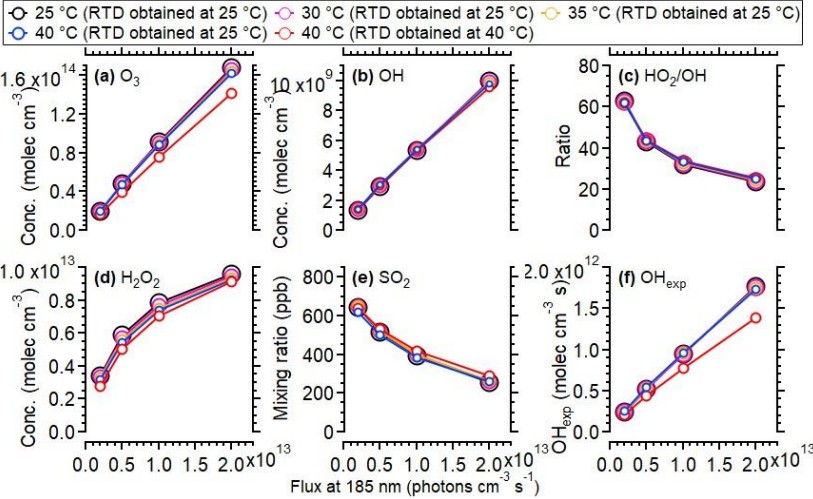

**Figure 9: Simulated concentrations of different oxidants from 25 to 40 °C in the OFR based on the radical mechanism of the KinSim model. 800 ppb of initial $SO_2$ and 2.2% water vapor mixing ratio (25 °C, 70% RH) were used. The simulated results using measured RTD at 40 °C are also shown.**

### 3.4 Temperature influence on SOA formation

The temperature can influence the SOA formation through changing gas/particle partitioning and $RO_2$ fate, thus affecting the SOA yield and chemical composition (Takekawa et al., 2003; Li et al., 2007; Zhang et al., 2015; Price et al., 2016; Quéléver et al., 2019; Kristensen et al., 2020; Atkinson et al., 1987). Here, we estimated the OA losses based on literature results and modeling work.

For the existing OA which is sampled into the OFR, the evaporation of input OA upon heating can be roughly estimated based on the results of thermal denuder (TD) experiments, in which the aerosols were heated from ambient/room temperatures (20-25 °C) to higher temperatures (typically 60-200 °C) at a typical residence time of 10-60 s (typically ~20 s) (Huffman et al., 2009; Xu et al., 2020; Hu et al., 2016; Saha et al., 2017; Kolesar et al., 2015; Saha and Grieshop, 2016; Lee et al., 2011). Previous studies suggest the evaporation of aerosol is kinetically limited, which varies with temperature, residence time, OA mass and volatility, phase state and dissociation rates for oligomers (Cappa, 2010; Riipinen et al., 2010; Roldin et al., 2014; Schobesberger et al., 2018). Based on the previous TD studies, the evaporation rate of $1\times10^{-4}$-$2.5\times10^{-4}$ per (°C·second) was usually observed under 50 °C of TD for ambient OA (typical average mass concentration: 10-30 μg $m^{-3}$) (Huffman et al., 2009; Feng et al., 2023; Paciga et al., 2016). In an OFR experiment conducted on an aircraft, Nault et al. (2018) found an average mass loss of ambient OA is ~32% due to enhanced temperature in a dark OFR compared to the unperturbed air sampled (+17 °C on average) at a residence time of 150 s. That OFR experiment showed an evaporation rate of $1.2\times10^{-4}$ per (°C·second), which is within the range of reported values obtained with the higher temperature TD experiments ($1\times10^{-4}$-$2.5\times10^{-4}$ per (°C·second)). Theoretically, when the temperature enhancement was assumed to be ~5 °C and a residence time of 160 s was applied, a mass loss of 8-20% was estimated for ambient OA in OFR. The enhanced



temperature inside of OFR might also impact other semi-volatile inorganic species, e.g., ammonium nitrate (Heim et al., 2020).

For the newly formed SOA in the OFR, the temperature impact was simulated based on the SOM model (He et al., 2022; Eluri et al., 2018). Specifically, the kinetic phase partitioning of SOA generated via OH-oxidation of typical ambient VOCs was simulated (Figs. 10-11, S12-17) as a function of OFR temperature ranging from 20-40 °C (binned in 5 °C) and OA mass concentration from 1 to 80 µg m$^{-3}$. Note that the results including SOA yield, size distribution, and O:C ratio for n-dodecane (Figs. 10 and 11), α-pinene (Figs. S12 and S15), toluene (Figs. S13 and S16) and m-xylene (Figs. S14 and S17) were examined under both high NOx and low NOx conditions. For dodecane, the simulated temperature-dependent SOA under high NOx in the SOM model generally is consistent with the chamber results obtained in Lamkaddam et al. (2017).

Generally, the higher temperatures result in lower SOA yields due to the increased evaporation of gas-phase products (Hildebrandt et al., 2009; Warren et al., 2009; Qi et al., 2010; Denjean et al., 2015). As shown in Figs. 10a and S12-S14, when RTD at 25 °C was used for different temperatures, the SOA yield of different VOC species including dodecane, α-pinene, toluene, and xylene can decrease by ~ 20% for typical temperature enhancement of 5 °C in the OFR, and by up to 40-50% at 40 °C compared to the values at typical 25 °C under high NOx condition. This decrease in SOA yield signifies the substantial influences of temperature on SOA formation inside of OFR due to heating. The simulated particle size also shows decreases as the OFR temperature goes up. The higher SOA mass and larger size of particles formed at lower temperatures are consistent with more gas-to-particle phase partitioning, and have also been observed in various temperature-controlled chamber studies (Clark et al., 2016; Lamkaddam et al., 2017; Boyd et al., 2017; Gao et al., 2022; Pathak et al., 2007; Tillmann et al., 2010; Price et al., 2016; Kristensen et al., 2020). When the measured RTD at 40 °C was applied in the model, an even lower SOA yield was achieved due to the shorter residence time of reactants. Under low NOx conditions, a smaller reduction in size distribution (Fig. 11 and Figs. S15-S17) and SOA yield, which are 5-10% for temperature enhancement of 5 °C and up to 15-35% for 15 °C as shown in Fig. 10 and Figs. S12-S14) was found compared with high NOx conditions. The smaller reduction was mainly due to more SOA with lower volatility being formed under low NOx conditions (e.g., acids, hydroperoxide) than the high NOx conditions (Kroll and Seinfeld, 2008; Srivastava et al., 2022; Presto et al., 2005; Aruffo et al., 2022). The parameters of the SOM model were obtained based on fitting the results to the chamber results under high and low NOx conditions, respectively (Eluri et al., 2018; Cappa et al., 2016), thus, the parameters were set such that the volatility of SOA decreases larger as a function of the functional groups added under low NOx condition than the high NOx condition (Cappa et al., 2013). Note that in the model we did not specifically treat the RO$_2$ fate e.g., a varied fraction of highly oxygenated organic molecules from autoxidation under different temperatures. Using a constant yield for HOMs in the model might lead to a lower HOMs mass fraction in total SOA under high temperatures, which would result in an underestimation of the oxidation level and an overestimation of the volatility of SOA formed under low NOx conditions (Bianchi et al., 2019). In addition, the wall loss was corrected in the SOM model with a constant wall loss rate of 2.5x10$^{-3}$ s$^{-1}$, which has been verified in the comparison of SOA model work between OFR and chamber studies (He et al., 2022). In theory, when the temperature of the OFR tube increases, the wall loss rate of low volatile organic compounds shall be smaller than under ambient temperature. Therefore, using a constant wall loss rate here might underestimate the SOA yield.

In previous studies, to correct for heating effects on SOA yield, Chen et al. (2013) and Lambe et al. (2015b) corrected SOA yield by 2% per K of temperature rise relative to 298 K (Stanier et al., 2008). However, this is a rough correction, because the SOA yields of different precursors and reaction conditions are affected by temperature by different amounts. For example, 0.41-0.52% per K was found for dodecane under high NOx conditions based on the SOM model while 0.87-





0.89% per K was found for α-pinene (Table S4). Note that in the OFR experiments, the decrease of SOA yield might be even larger due to the potential mass loss of seed OA upon heating in the OFR. The simulated results from the SOM model here provide an approximate reference to help recalibrate the SOA formation inside of OFRs. As the detailed numbers

calculated by the SOM model for different species under high and low NOx conditions can be seen in Table S4. For a specific recalibration, e.g., mixed precursors, a more detailed model or temperature-controlled experiments considering the mixing effect of precursors can be performed (Mcfiggans et al., 2019).

For the SOA chemical composition, the higher temperature inside the OFR results in increased O:C ratios of SOA, e.g., ~0.35 (25 °C) vs ~0.42 (40 °C) in dodecane experiments under high NOx conditions and ~0.23 (25 °C) vs ~0.28

(40 °C) under low NOx condition. The enhanced O:C ratio under higher temperatures was also found in the chamber results for m-xylene/NOx photooxidation in Qi et al. (2010) and $O_3$ oxidation of α-pinene in Denjean et al. (2015). The higher O:C under higher temperatures was probably caused by the evaporation of semi-volatile and less-oxidized components with increased temperatures (Clark et al., 2016; Gao et al., 2022).

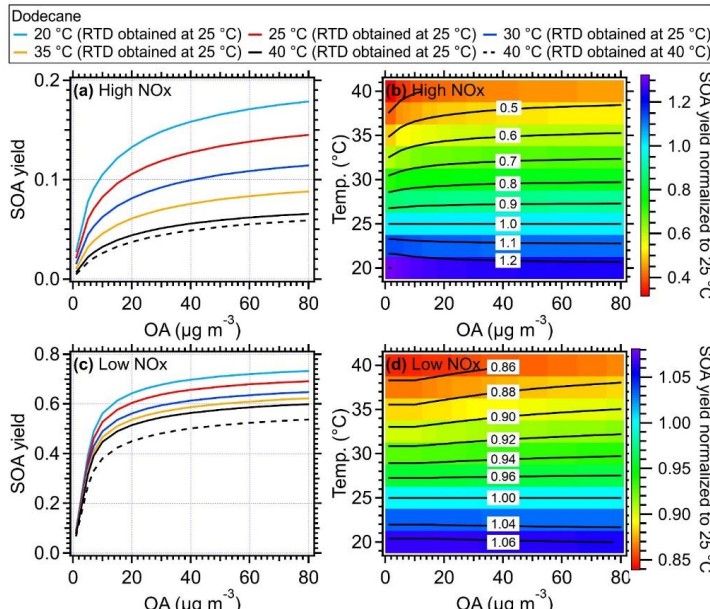

**Figure 10: Simulated SOA yield of dodecane as a function of mass concentration of organic aerosol and temperature inside OFR for (a) high NOx and (c) low NOx conditions, respectively. The simulated results using the measured RTD obtained at 40 °C are shown as black dashed lines. The ratio of SOA yield of dodecane from different temperatures compared to that of 25 °C under (b) high NOx and (d) low NOx conditions. The equivalent aging time is 1 day by assuming the OH concentration is equivalent to 1.5×10$^6$ molecule cm$^{-3}$ (Mao et al., 2009).**

In the model, the evaporated SOA mass as a function of temperature was mainly determined by the enthalpy ($H_i^{vap}$), as described by the Clausius–Clapeyron Equation (E.q. (1)). A varied enthalpy $H_i^{vap}$ dependent on saturation concentration (C*, $H_i^{vap}$=-11×$logC_{ref}^*$) was applied as default setting based on Epstein et al. (2010). Various experimental studies using thermodenuder for measuring OA volatility show the ambient $H_i^{vap}$ for OA might vary in a range of 50-150 kJ mol$^{-1}$ (Epstein et al., 2010; Saha et al., 2017; Cappa and Jimenez, 2010). To further explore the sensitivity of temperature-

dependent SOA formation toward enthalpy, we demonstrate the SOA yield as a function of temperature under constant

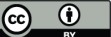



$H_i^{vap}$ values in Fig. 12 and S18. The results suggest SOA yield obtained with varied $H_i^{vap}$ is generally similar to that when a constant $H_i^{vap}$ of 100 kJ mol⁻¹ was applied, which is the most commonly reported $H_i^{vap}$ for ambient OA. Higher $H_i^{vap}$ suggests the SOA formation is more sensitive to the temperatures. When the upper (150 kJ mol⁻¹) and lower limit (50 kJ mol⁻¹) of $H_i^{vap}$ are applied, the simulated results of SOA yield suggest a maximum of 22% and 90% variation, respectively,

under high NOx condition and 18% and 42% variation, respectively, for low NOx condition based on different species and temperature.

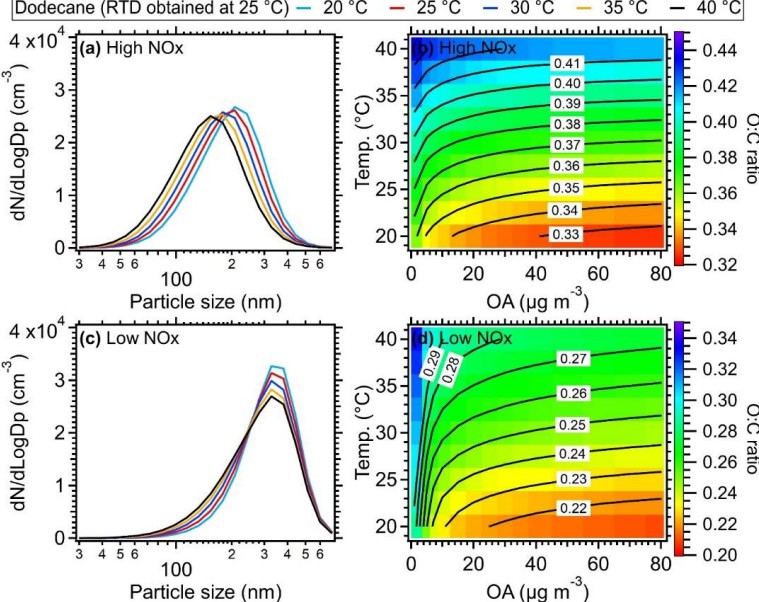

**Figure 11: Size distribution of dodecane under different temperatures and 30 μg m⁻³ organic aerosol for (a) high NOx and (c) low NOx conditions, respectively. The O:C ratio of dodecane under different temperatures and organic**

**aerosol concentration under (b) high and (d) low NOx conditions. The equivalent aging time is 1 day by assuming the OH concentration is equivalent to 1.5×10⁶ molecule cm⁻³ (Mao et al., 2009).**

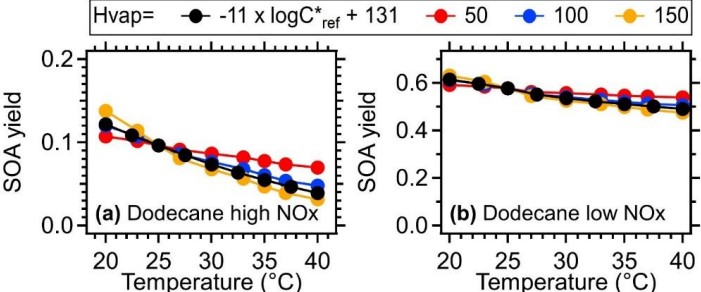

**Figure 12: SOA yield of dodecane as a function of temperature under different $H_i^{vap}$ values (Unit: kJ mol⁻¹). The mass concentration of organic aerosol is assumed to be 15 μg m⁻³. The equivalent aging time is 1 day by assuming**

**the OH concentration is equivalent to 1.5×10⁶ molecule cm⁻³ (Mao et al., 2009).**





In summary, the heating effect introduced by the lamps can have an important impact on the SOA formation inside of OFR for certain high-OH exposure applications, which shows varied extents for different precursors and reaction conditions. The simulated results suggest that the deceased ratio of SOA output vs preexisting OA at higher OH exposures (e.g., equivalent aging time >~5 days under low NOx conditions) in previous experiments (Hu et al., 2022; Lambe et al., 2015a; Ortega et al., 2016; Palm et al., 2016; Saha et al., 2018) might not only due to heterogeneous reaction and/or enhanced gas-phase reaction, but also probably caused by the heating evaporation. However, the yield and ambient OFR results under equivalent aging time <~4 days shall still be valid (Palm et al., 2018). The enhanced temperature might also impact the phase of aerosols by changing the chemical composition of OA (viscosity, O:C, etc.), as the OFR temperature covers the usual range of glass transition temperature of ambient OA (2-87 °C) (Li et al., 2020b; Derieux et al., 2018). The impact of enhanced temperature on the phase state (i.e., viscosity) of aerosol phases, wall loss or other effects needs to be further studied.

**3.5 Approaches to reduce the heating effect**

In the OFR, one approach to reducing the heating influence of the lamps is to decrease the OFR residence time, at the expense of decreasing the maximum achievable OH exposure and time available for LVOC condensation onto aerosols (Peng and Jimenez (2020). Another approach involves using fewer lamps at lower voltage settings; for example, using two lamps at less than 3 V maintains a temperature enhancement of less than 5 °C while still achieving OH exposures of up to 5 days under low NOx conditions. In addition, it can also be useful to increase heat transfer away from the OFR, to reduce its operating temperature e.g., blowing air with fans or air conditioners. To test this, we used two big fans (45 cm in diameter at a distance of 30 cm from the OFR) pointed to the lower parts of the OFR. We found the maximum delta temperature (OFR minus ambient air) for a typical high light setting decreased from 16 °C to 7 °C (44% deduction), as shown in Fig. 13. That test showed that blowing the flow tube with fans is a very efficient way to offset the lamps heating, and is simple with no major trade-offs. Designing a cooling system on the outer surface of OFR by using circulating water or cold air might also be a good way to better control the temperature inside of OFR (Watne et al., 2018; Xu and Collins, 2021; Huang et al., 2017; Liu et al., 2018; Chu et al., 2016; Zhao et al., 2021), however, it would require a substantial redesign of the hardware of OFR tubes and beyond the scope of this manuscript.

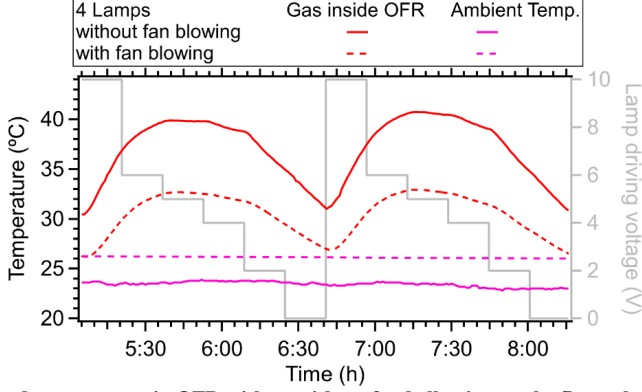

**Figure 13: The measured temperature in OFR with or without fan bellowing on the flow tube in the laboratory. The ambient temperature was also shown. Two fans were used, with a diameter of 45 cm and a rotation rate of 1400 r min[-1].**



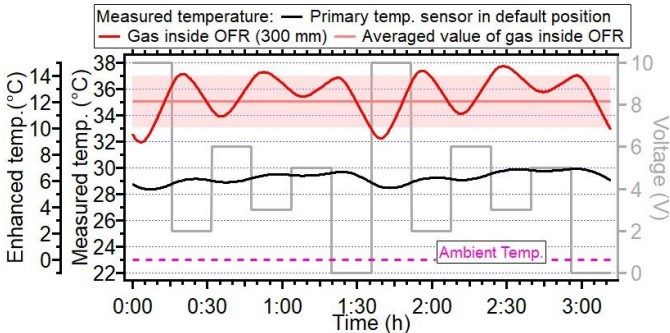


**Figure 14: The measured temperature variation when lamp driving voltage cycles were set in a non-monotonic pattern (10-2-6-3-5-0 V for one cycle). The red line and shading are the average temperature and standard deviation (35.05±1.97 °C) measured at the centerline (probing distance: 300 mm) inside OFR. The purple dotted lines represent the ambient temperature. Each lamp voltage was set for 16 min.**

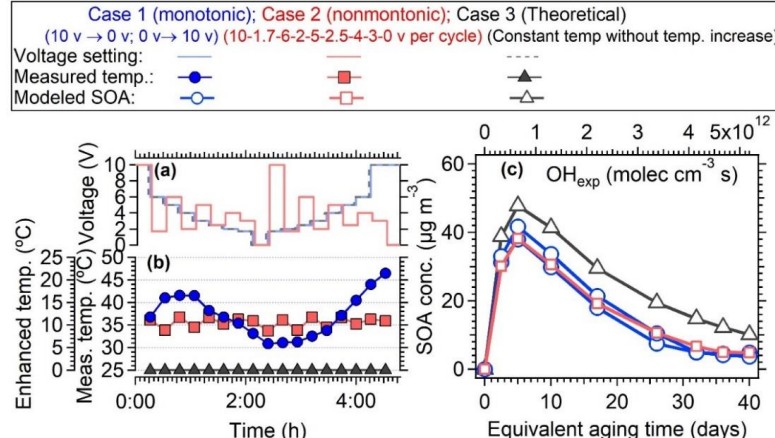


**Figure 15: (a, b) The temperature variation with lamp driving voltage is explored using a monotone (10→0→10V) and non-monotone pattern lamp power change pattern, respectively. (c) the modeled SOA formation from oxidation of 10 ppb toluene (OM=30 μg m⁻³) with OH radicals as a function of OH exposure. The SOA formation under monotone and non-monotone scenarios are both simulated. The case 3 of "theoretical" indicates the temperature**

**were set to be constant (i.e., 25 °C).**

The voltage setting strategy in the OFR also can be improved. In most of the OFR laboratory and field studies, OFR users usually changed the light setting monotonously from 0 to 10 V or from 10 to 0 V, which results in a continuously changing temperature inside of OFR (Fig. 2, Fig. 5 and Fig. S4), therefore resulting in a variable SOA formation yield under different temperatures (Fig. 10). To mitigate the variable heating effect, we suggest alternating between high and

low voltage settings to minimize the heat accumulation. Fig. 14 shows an example of such a light setting cycle using 10-2-6-3-5-0 V. In such a way, although the average temperature inside of OFR is still higher than the ambient temperature, the variation of measured temperature in the OFR can be kept within a narrow range and has a smaller temperature deviation (±1.97 °C) compared to the usual settings (Fig. 14a, ±4.76 °C). Another advantage is that in such a way, the variation trend of SOA yield as a function of OH exposure would have fewer uncertainties compared to those performed under





monotonically increasing or decreasing light settings. As shown in Fig. 15c, when using the measured temperature to simulate the SOA formation from toluene in the OFR, the SOA showed two different trends (2-32% differences) when monotonic and non-monotonic light settings cycles were applied. The SOA formed from the proposed non-monotonic lamp setting cycle shows better agreement on the variation of mass concentration (Fig. 15c) and mass ratio (Fig. S19) compared with the SOA formation at 25 °C (40.4-80.0% for non-monotone setting vs 32.9-87.3% for monotone setting), which would

introduce fewer uncertainties in the SOA yield and other aspects, e.g., chemical composition.

**4 Conclusion**

We systematically measured the temperature distribution inside of the lamp-enclosed Aerodyne PAM oxidation flow reactor. It is found that the lamps inside of OFR create heating energy, thus leading to temperature increase inside of OFR. A brief summary of the temperature increase and its influencing factors can be found in Table A1. The enhanced air

temperature increases as a function of the setting of lamp voltage (OH exposures) and the number of lamps applied, decreases as a function of flow rate due to a shorter residence time at higher flows. The distribution of enhanced temperature varied with special position due to the complex effects of thermal transfer and flow mixing. With default lamps from Light Source Inc. installed, the temperature increase of air in the OFR is generally below 5 °C (at central line) when the driving voltage of two lamps is below 3 V (typically < 5 days of equivalent atmospheric OH exposure under low-NOx conditions)

and the flow rate is 5 L min$^{-1}$ (average residence time, $\tau_{avg}$=~150-180 s). The usage of BHK lamps typically yields smaller temperature enhancement in OFR at the same OH$_{exp}$ due to its lower power dissipation. The heating energy loss of the OFR system is mainly through the walls of the reaction chamber, then by exiting air inside of OFR and/or purged nitrogen.

The impact of the enhanced temperature on the flow distribution, gas, and aerosol phase chemistry inside of OFR was systematically evaluated. The pulsed tracer measurement results suggest the enhanced temperature in OFR accelerates the

diffusion, thus resulting in a shorter average residence time ($\tau_{avg}$). Although box model simulation results show that the temperature increase in OFR has a negligible impact on gas-phase oxidant concentrations (<~5%), however, has a certain impact on the $\tau_{avg}$ related parameters, e.g., OH exposure (~20% decreases results from temperature increase of 15 °C).

The increase of temperature has larger impacts on the aerosol phase chemistry than on the gas phase. When the enhanced temperature inside the OFR is around 5 °C at a flow rate of 5 L min$^{-1}$, the evaporation loss of ambient OA was

found to be 8-20% based on thermodenuder and field experiments. The simulation from the SOM model suggests the SOA yield from four typical precursors (n-dodecane, α-pinene, toluene, and m-xylene) can decrease <20% under high NOx conditions and <10% under low NOx conditions when the temperature in OFR increases 5 °C. With the increase of temperature, the size distributions also show a substantial decrease while increases in the O:C ratios were found. This work demonstrates the substantial influence of temperature on SOA formation and highlights the need for consideration of the

temperature impact when using OFR as a tool for investigating aerosol chemistry. Based on the SOM model simulation results in Table S4, we recommend 0.19-1.6% and 0.26-3% per K for temperature effect correction for SOA yield from four typical precursors under high NOx and low NOx conditions, respectively.

In general, applying higher flow rate and lower lamp power, cooling the reactor with fans are recommended to keep the temperature enhancement low in the OFR system. In addition, to control the variation of enhanced temperature

introduced by the monotonic trend for lamp settings (e.g., 0→10 V or 10→0 V), we propose to set the voltage with high and low voltage alternatively (e.g., 10-2-6-3-5-0 V) to improve the heating accumulation and keep the temperature variation within a narrow range. In summary, our evaluation of the temperature enhancement inside of lamp-enclosed OFR helps to





improve our understanding of flow distribution and chemistry inside of this OFR type, which can help to reduce the uncertainty of OFR usage in the future.

Appendix

**Table A1: Summary on the effect of different dimensional factors on the temperature of air inside of OFR, as well as the impact of temperature enhancement on flow, gas and aerosol chemistry inside OFR.**

| Factors influence on temperature (T) of air inside OFR | | |
|---|---|---|
| | **Impact factors** | **Effect** |
| Heating energy input | Driving voltage of lamps | Driving voltage↑ → T↑ |
| | Number of lamps | NO. of lamp used ↑ → T↑ |
| | Lamp types | No differences for the same brand |
| | Lamps lasting time | Time↑ → T↑ before balance |
| | Setting voltage sequence | Set voltage ↑ monotonically → T↑; Set voltage ↓ monotonically → usually delayed T ↑ peak |
| Heating energy loss | Flow rate (corresponds to Residence time distribution, RTD) | Flow rate ↑ (RTD ↓) →T↓ |
| | $N_2$ purge air | Flow rate ↑ or $N_2$ temperature ↓ → T↓ |
| | Surrounding T of OFR | Surrounding T↓ → T↓ |
| | Metal tube temperature | Metal tube T↓ → T↓ |
| Temperature distribution | Measured position | Closer to lamp sleeve → T↑ |
| | *Vertical* | Generally, T ↑ in the upper position |
| | *Horizontal* | Symmetrically distributed. |
| | *Probing depth* | Probing depth↑→T↑ for ARI OFR; Probing depth↑→T↓ for Penn State OFR |
| Temperature influences on flow, gas, and aerosol chemistry | | |
| | **Factors being influenced** | **Effect** |
| Flow | Average RTD | T↑ → RTD ↓ |
| Gas-phase | absolute concentration of oxidants | T↑ → Minor impact (<5%) |
| | OH exposures | T↑→ OH exposure ↓ |
| | HOM yield | T↑→ HOM yield ↑ |
| Aerosol-phase | input/seed OA | T↑→ OA masses ↓ |
| | SOA yield* | T↑→ SOA yield ↓ |
| | Size distribution of SOA* | T↑→ Peak sizes ↓ |
| | Oxidation level of SOA* | T↑→ Oxidation level ↓ |

*More impact on high NOx regime than low NOx regime.

*Data availability.* The data shown in the paper are available on request from the corresponding author
(weiweihu@gig.ac.cn)

*Author contributions.* HW, AL, QH, YH, MM, SJ, YZ, ZP, YH, DD, PJ, and JJ: conceptualization, methodology, investigation and revision. MH: simulation of flow in OFR. QH and HC: provide the solid work design of OFR. WH and XM: supervision, project administration and funding acquisition. TP and WH wrote the paper, and all co-authors supported the interpretation of the results and contributed to improving the paper.

*Competing interests.* The authors declare that they have no conflict of interest.



*Acknowledgement*:This work was supported by the National Key R&D Program of China, Young Scientist Program (No. 2021YFA1601800), the National Natural Science Foundation of China (grant No. 42375105, 42275103, 42230701), Guangdong Pearl River Talents Program (2019QN01L948), Foundation for Program of Science and Technology Research (2023B1212060049). ZP, MM, PJ, DD and JJ were supported by NSF Atmospheric Chemistry (AGS 2206655).

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
