# Peer review of "A comprehensive evaluation of enhanced temperature influence on gas and aerosol chemistry in the lamp-enclosed oxidation flow reactor (OFR) system"

_Atmospheric Measurement Techniques, 2023_

## Referee Comment (RC3)

**Review of "A comprehensive evaluation of enhanced temperature influence on gas and aerosol chemistry in the lamp-enclosed oxidation flow reactor (OFR) system"**

Tianle Pan et al. present a study on the influence of the heating effect of oxidation flow reactor (OFR) UV lamps on oxidant chemistry, flow conditions and secondary organic aerosol (SOA) formation. The authors find that while the increase in temperature does not greatly affect the oxidant chemistry, the effect of temperature gradient on residence time distribution shortens the mean residence time. Thus, the OH exposure at higher light intensities is affected by the heating effect. The temperature increase in the OFR influences the SOA yields and composition, so that the SOA yield is lower at higher temperatures and the O to C ratio of SOA is higher at higher temperatures.

The heat caused by the OFR lamps is an important aspect of OFR studies that has not been well characterized earlier. The measurements and analysis in this study are comprehensive and thoroughly done. However, the quality of reporting and language is not fully adequate. Another concern is the relevance of the SOM model for the SOA formation in the OFR. Thus, I recommend publishing this study in Atmospheric Measurement Techniques, but only after the following remarks have been addressed. Especially the language needs to be improved.

**General comments**

1. The reporting should be concise, and all statements should be well-defined and unambiguous. Currently, the text contains unnecessary extra words (like 'around', 'might'), which are sometimes necessary but could be mostly removed. The authors often use word 'support', which in many places could be replaced by a more exact wording. Furthermore, for many sentences in Section 3 it is unclear what the authors actually mean. Some of these are included in the following specific comments, but I recommend the authors read through the paper and for each sentence ponder whether the sentence is necessary, and whether the statement is unambiguous.

2. It is unclear why the authors used the ring flow manifold in RTD measurements when it was not used in the simulations. In my opinion, this is justified since the ring flow is typically used in actual measurements. However, the authors could discuss this justification when they describe the RTD measurement and simulation. This difference between measurement and simulation should also be mentioned in Fig. 8 caption. In addition, I think the description of RTD measurement (p. 13, l. 412-415) should be moved to Section 2. What were the flow rates in aerosol line and exhaust line during RTD or temperature measurements? The authors provide the flow rates in different occasions, but it is not clear whether this is the total flow rate (aerosol line + exhaust line) and what is the ratio between the aerosol line and exhaust line.

3. The authors use term 'Enhanced temp.' in figures to describe the difference between OFR and ambient temperature. In my opinion, e.g. 'Temp. enhancement' or just $\Delta T$ would be better.

4. When discussing the approaches to reduce the heating effect in Sect. 3.5, the authors actually only discuss approaches to reduce the effect of increased temperature. For example, when the fans are used to cool down the OFR external surface, the heat transfer inside the OFR is improved and this affects the RTD since the heat transfer occurs via convection. However, the authors did not characterize the effect of different cooling methods on the RTD, which at least should be mentioned in this section or the section headline should be changed.

5. The authors sometimes mix past and present tense. See e.g. Fig. 3 caption: four lamps were turned on, flow rate is 5 lpm.

6. Based on the current description of the SOM model, it seems that the authors first model the SOA formation in "normal" temperature and then study its evaporation in elevated temperature (e.g. p. 17, l. 494: "for the newly formed SOA in the OFR, the temperature impact was simulated based on SOM model"). In that case, what is the temperature where the SOA formation is modeled? If this is the case, the model results regarding e.g. the SOA yield are not very relevant, because in the OFR the SOA formation would take place in the elevated temperature.

   The authors should change the heading of Section 3.4 to "Temperature influence on OA evaporation", and describe more clearly that they are modeling the evaporation of SOA that was formed in temperature X and then injected into the OFR at elevated temperature. This needs rewriting of Sect. 3.4 so that the authors discuss OA evaporation instead of SOA formation.

**Specific comments**

p. 1 l. 44: box model using radical chemistry → radical chemistry box model

p. 2 l. 58: or → for

p. 2 l. 82: The high temperature inside the OFR does not cause the recirculating flows, it is the temperature gradient caused by lamp heating.

p. 3 l. 88: SOA simulation and study → SOA simulations and studies

p.3 l. 102: The acronym ARI has not been declared earlier.

p. 4, Fig. 1: What is the exhaust gas line? It seems that sensor (3) is measuring the ring flow outlet, but in the text it is unclear whether the exhaust gas means the ring flow or the N2 purge flow. Please clarify.

p. 7 Fig. 2 caption: "assuming ambient OH concentrations are around 1.5e6..." I suppose you have used an exact value of 1.5e6 in the calculations, so please remove the word 'around' (also earlier regarding the water mixing ratio).

p. 10 l. 296: "In our case, the..." → "In our case, when the..."

p. 10 l. 298: What is "vertical axial direction"? I think axial would mean the direction of the central axis of rotation. It would be helpful to define the different directional terms (vertical, horizontal, probing depth) graphically e.g. by adding another panel in Fig. 1.

p. 11 l. 319: "OFR chambers (OFR metal tubes)" → "OFR  surface"

p. 11 l. 331: surfaces → surface

p. 11 l. 333-339: The authors discuss in length why the temperature of the external surface is lower than that of the gas inside the reactor, while it is basic physics that since the OFR is not isolated and the ambient temperature is lower than the OFR internal temperature, there will be heat transfer from inside the OFR to the ambient, and the medium between these (the surface) will be at lower temperature than the OFR internals. This is correctly described in lines 338-339. The thermal mass of the OFR is not relevant, as it affects only the rate of temperature increase but not the final equilibrium temperature that is discussed here.

p. 12 l. 352: Here and elsewhere the authors use term 'OFR tubes' for the OFR casing. I would suggest finding a better term, since to my understanding this structure consists of a single large tube and the end plates, not 'tubes'.

p. 13 l. 395: was → is

p. 13 l. 396: What is 'more even and better-mixed air'? I think the purpose of the ring flow manifold is to reduce dead space at the end of the OFR and thus reduce the mixing.

p. 13 l. 397: Please check the tense of this sentence. Also, the flow distribution certainly will be different, so the word 'might' is not necessary here.

p. 13 l. 409-411: The definition of "sum of integrals of signal over time versus the sum of signals" is not correct. Please define the mean residence time correctly or just cite the relevant source.

p. 13 l. 412: Here and elsewhere: the authors have measured RTDs and thus also the mean residence times. "Average RTD values" is not correct term.

p. 14 l. 427: calculation → simulation ?

p. 14 Fig. 7:

- In panels a, c and e, there are 4 different cases in each panel. I assume these represent the situation at probing depths of 100, 200, 300 and 400 mm. Please indicate this somehow in the figure or in the caption.
- In panels g-j: What does the horizontal or vertical velocity mean? If these are the velocities towards the horizontal or vertical direction, then the labels 'backward' and 'forward' do not make sense. Or should it be axial velocity?
- Figure caption: "Enhanced temperature influences in panels (c) and (d) were considered" – should this be panels (e) and (f)? The last reference to panel (h) should be (j).

p. 15 Fig. 8: The y axis label should be "Probability density function" (or Residence time distribution) with unit of $s^{-1}$, not "probability distribution function". In figure caption, "The average RTD values were also estimated here" → "The mean (average) residence times are also shown here". The calculation of the mean residence time is not an estimate, but a well-defined property of the RTD.

p. 15 l. 456: RTD distributions → RTDs

p. 15 l. 460: reaction rate**s**, concentration**s** of

p. 16 l. 467: "very minor to negligible impact on the gas-phase reactions". This is too strong a statement since only a limited set of gases and their reaction pathways was studied.

p. 16 l. 485: per (°C second) → $s^{-1}°C^{-1}$

p. 16 l. 487: mass loss of ~32% for ambient OA

p. 17 l. 492: "might" is unnecessary here

p. 17 l. 519-520: unclear sentence

p. 17 l. 521: "Using a constant yield for HOMs..." → "The constant yield for HOMs used in this model might..."

p. 18 l. 534: "As the detailed..." → "The detailed..."

p. 19 l. 557-558: Unclear sentence. Consider e.g. "Increasing Hvap increases the sensitivity of SOA formation to temperature".

p. 19 l. 561: what does "based on different species and temperature" mean?

p. 19 Fig. 11: Is the 30 µg/m3 the inlet concentration? And is the x-axis in panels b and d the inlet concentration of OA?

p. 19 Fig. 12: "the ambient OH concentration"

p. 20 l. 572-573: "which shows varied extents for different precursors and reaction conditions" – what does this mean?

p. 20 l. 573: deceased → decreased

p. 20 l. 583: reducing → reduce

p. 20 l. 590: deduction → reduction

p. 20 Fig. 13: bellowing → blowing

p. 22 l. 621: trends → hystereses ?

p. 22 l. 639-640: is the shorter residence time really due to accelerated diffusion?

---

## Author Comment (AC2)

**Response to reviewers for the paper "A comprehensive evaluation of enhanced temperature influence on gas and aerosol chemistry in the lamp-enclosed oxidation flow reactor (OFR) system" in Atmos. Meas. Tech. Discuss. Doi: 10.5194/amt-2023-230**

**By Tianle Pan et al.**

We appreciate the three reviewers' comments and support for the publication of this manuscript after revisions. Following the reviewers' suggestions, we have carefully revised the manuscript. To facilitate the review process, we have copied the reviewer's comments in black text. Our responses are in regular blue font. We have responded to all the referee comments and made alterations to our paper (**in bold text**).

**Anonymous Referee #2**

**General Comments**

Pan et al. observed a lamp-induced enhanced temperature inside the PAM-OFR based on measurements and investigated the impacts on flow and chemistry using model simulations. They find that the temperature enhancements have negligible impacts on gas-phase reactions, while large impacts on the SOA yields, chemical composition, and aerosol-phase chemistry. This study provides relatively systematically and detailed heating effects on chemistry inside the PAM-OFR, and should be suitable for publication in AMT. However, I have a few concerns that I would like the authors to address and some suggestions for improving the clarity of presentation.

**Major Specific Comments**

**R2.1:** The authors use "PAM-OFR" in the introduction and methods sections, while "OFR" is used in the rest of the manuscript. Can the authors use one terminology to keep constant throughout the manuscript?

**A2.1** We replaced the "OFR" with "PAM-OFR" in the manuscript when the PAM-OFR

was specifically referred to.

**R2.2:** The authors find shorter residence time under the enhanced temperature than non-heated PAM-OFR. Generally, shorter reaction time leads to lower SOA yields. How would this contribute to lower SOA yields in SOM modeling results? Compared to gas-phase products evaporation, which is more important?

**A2.2:** This is a really good question. A short answer is the evaporation under high temperature impact more SOA formation in OFR than the change of residence time. As shown in Figs.10a and 10c, when the SOM results were calculated based on the residence time distribution (RTD) obtained at 25 °C, the SOA yield of dodecane decreased ~60% under high NOx condition and ~14% under low NOx condition as the temperature inside PAM-OFR increased from 25 °C (the red line) to 40 °C (the black line). When we considered the really measured RTD obtained at 40 °C (the black dashed line), the SOA yield of dodecane was even lower. But the impact of the RTD was weaker than the temperature, as only ~8% (high NOx) and ~10% (low NOx) decrease in the SOA yield was found compared to the results with RTD obtained at 25 °C (the black line). We revised the sentences related to the RTD on SOA formation as shown in the following:

Line 535-539: "**When the measured RTD at 40 °C was applied in the model, an even lower SOA yield was achieved due to the shorter residence time of reactants. However, this influence was weaker than the directly influences caused by the temperature increase on SOA formation. A decreased of ~8% of dodecane SOA yield was found at 40 °C under high NOx condition and 10% under low NOx compared to the results with RTD measured at 25 °C (Fig. 10).**"

[Figure]

**Figure 10: Simulated SOA yield of dodecane as a function of mass concentration of organic aerosol and temperature within the OFR under (a) high NOx and (c) low NOx conditions, respectively. The simulated results using the measured RTD obtained at 40 °C were shown as black dashed lines. The ratio of SOA yield of dodecane from different temperatures compared to that of 25 °C under (b) high NOx and (d) low NOx conditions. The equivalent aging time was 1 day by assuming the OH concentration equated to $1.5 \times 10^6$ molecule cm$^{-3}$ (Mao et al., 2009).**

**R2.3:** For low and high NOx conditions, what are the concentration levels? If the authors intend to distinguish the fate of peroxy radicals in two conditions, NO concentrations should also be provided.

**A2.3:** In the PAM-OFR experiments, we distinguish high and low NOx conditions by the ratios of $RO_2$ reacted with NO or $HO_2$. Based on the definition in the most experiments studies, $r(RO_2+NO)/r(RO_2+HO_2) > 1$ represents the high NOx condition (Peng and Jimenez, 2017). However, in the SOM model, the reaction under low/high NOx conditions was processed by using different parameters ($m_{frag}$, $\Delta LVP$, p1-p4, Table

S3) without considering concentration of NO gases in PAM. These model parameters were obtained by fitting the simulated results to the measured chamber results under high and low NOx conditions, respectively (Eluri et al., 2018; Cappa et al., 2016).

We added the corresponding explanations in line 226-228: "**These parameters were obtained by fitting the simulated results to the measured chamber results under high and low NOx conditions, respectively (Eluri et al., 2018; Cappa et al., 2016).Thus, the exact NO concentration was not considered in the SOM model during the simulation**"

**R2.4:** It would be helpful if the authors can provide experiment results to show the heating effects on SOA formation inside the PAM-OFR, e.g. using *vs* not using the external fans.

A2.4 Thanks for your suggestions. We added the measured results of SOA formation in PAM-OFR using vs. not using fans. Distinguished decrease on the SOA masses was found when the fans was not used, confirming the heating effect on SOA formation inside the PAM-OFR, as shown in the following:

[Figure]

**Figure S19. The SOA formation from benzene and OH radicals in the PAM-OFR as a function of light intensity. Two cases including PAM-OFR was blown with fans and without fans were both shown. The room temperature and temperature**

measured with the primary Temp&RH sensor set in the back panel were shown in the right axis. Note the OFR temperature reported here is the lower limit as discussed in section 3.1.5. The gas-phase benzene (99.80%, Sigma-Aldrich) was generated with syringe pumps. Benzene was used as gas-phase precursor in this experiment due to its lower $k_{OH}$, since benzene will not be totally consumed under the high OH exposure at high voltage settings in OFR. The flow rate in this experiment was 4.5 L min$^{-1}$, and the RH was ~30%.

The explanation was also added in the revised miantext:

"To confirm the model results, we did a simple laboratory experiment and found the formed SOA masses was indeed substantially decreased in OFR due to the heating effect (Fig. S19), which is consistent with the simulated model results."

**R2.5:** Line 573: "decreased" not "deceased" I would say.

A2.5 We have corrected the typo.

**R2.6:** Line 585: one bracket is redundant.

A2.6 We have deleted the left bracket.

---

## Author Comment (AC4)

**Response to reviewers for the paper "A comprehensive evaluation of enhanced temperature influence on gas and aerosol chemistry in the lamp-enclosed oxidation flow reactor (OFR) system" in Atmos. Meas. Tech. Discuss. Doi: 10.5194/amt-2023-230**

**By Tianle Pan et al.**

We appreciate the three reviewers' comments and support for the publication of this manuscript after revisions. Following the reviewers' suggestions, we have carefully revised the manuscript. To facilitate the review process, we have copied the reviewer's comments in black text. Our responses are in regular blue font. We have responded to all the referee comments and made alterations to our paper (**in bold text**).

**Anonymous Referee #1**

**General Comments**

The manuscript by Pan et al. investigates the impact of lamp-induced heating in an Aerodyne PAM-OFR, assessing the temperature distribution, flow dynamics, and chemical consequences resulting from UV lamp heating. The authors have used CFD simulation, KinSim kinetic model, and SOM model to investigate how the temperature affects the flow and average OH exposure and how the enhanced temperature impacts the chemistry of gas-phase reactions and SOA formation. They find that the temperature enhancement can be up to 15 °C and it has impacts on the gas-phase chemistry and the yield, size, and oxidation levels of SOA. Overall, this manuscript gives a relatively comprehensive evaluation of the increased temperature on the chemical processes in the PAM-OFR. However, some concerns need to be addressed before the manuscript can be considered for publication in AMT.

**Major Specific Comments**

**R1.1:** The authors find that the heating inside PAM-OFR is mainly from the heat transfer of the hot quartz sleeve (heated by the lamps) but not from the optical radiation.

This is true since UV radiation generates little heat. Based on this finding, I would expect that the authors recommend moving the lamps out of the reactor, which can overcome the heating issue caused by the lamps. This can be found in the design of other OFRs in previous studies (e.g., Huang et al., Atmos. Meas. Tech., 10, 839–867, 2017; Simonen et al., Atmos. Meas. Tech., 10, 1519–1537, 2017; Li et al., Atmos. Chem. Phys., 19, 9715–9731, 2019) and should be discussed in "Section 3.5 Approaches to reduce the heating effect".

**A1.1:** We agree with that moving the lamps out of the reactor will help reduce the temperature increase. However, we do not think this method can overcome the heating issue. For examples, additional cooling methods were also applied for OFRs with UV lamps mounted outside. e.g., Huang et al. (2017) used circulating water to cool down the system; Li et al. (2019) used 30 L min$^{-1}$ N$_2$ through the quartz tubes; Four fans were used to dissipate the heat in Xu and Collins (2021); The temperature increase of the tube wall could be 8 °C inside the Go:PAM when the intensity of UV lamps was maximum and the fan was turned on (Watne et al., 2018).

Following the reviewer's suggestions, we declare that moving the UV lamps outside of OFR is a method to mitigate the heating issue.

"**Moving the UV lamps outside the tube and designing a cooling system on the outer surface of OFR with circulating water or cold air can also be effective ways to improve the temperature control inside of OFR (Watne et al., 2018; Xu and Collins, 2021; Huang et al., 2017; Liu et al., 2018; Chu et al., 2016; Zhao et al., 2021; Li et al., 2019), however, these will require a substantial redesign of the hardware of OFR tubes and are beyond the scope of this manuscript. And mounting the lamps outside of the OFR limits the use of OFR185 mode due to the low transmission efficiency of quartz glass for light at 185 nm (Simonen et al., 2017) and OFR254 mode is usually used**."

**R1.2:** The authors use the SOM model to investigate the influence of temperature on SOA formation, which highly relies on the performance of the model under different temperatures. It would be helpful to conduct SOA formation experiments with different temperatures to get accurate decreases in SOA yield under high temperatures. This

comparison can be done with or without efficient heat removal methods including a high volume of N2 purge air and external fans as the authors have shown in the manuscript.

**A1.2:** Thanks for reviewer's suggestions. We have incorporated these suggestions and did two experiments to prove that the SOA formation in the OFR was indeed decreased when the lights were on, as shown in Fig. S19 the revised manuscript. However, we cannot calculate the yield due to the PTR-MS which can be used to measure the VOCs mass concentration was broken in recent several months and still in repairing. Thus, we cannot compare the measured results with the SOM model simulation. More experiments will be done in the future.

[Figure]

**Figure S19. The SOA formation from benzene and OH radicals in the PAM-OFR as a function of light intensity. Two cases including PAM-OFR was blown with fans and without fans were both shown. The room temperature and temperature measured with the primary Temp&RH sensor set in the back panel were shown in the right axis. Note the OFR temperature reported here is the lower limit as discussed in section 3.1.5. The gas-phase benzene (99.80%, Sigma-Aldrich) was generated with syringe pumps. Benzene was used as gas-phase precursor in this experiment due to its lower $k_{OH}$, since benzene will not be totally consumed under the high OH exposure at high voltage settings in OFR. The flow rate in this experiment was 4.5 L min$^{-1}$, and the RH was ~30%.**

The explanation was also added in the revised mian text:

**"To confirm the model results, we did a simple laboratory experiment and found the formed SOA masses was indeed substantially decreased in OFR due to the heating effect (Fig. S19), which is consistent with the simulated model results."**

**R1.3:** Similarly, SOA formation experiments with different voltage setting strategies need to be added in Section 3.5 to show the effectiveness.

**A1.3:** The SOA formation experiment between benzene and OH radicals was done to prove the effectiveness of the cooling method. The detailed results can be found in A1.2.

**R1.4:** The high temperature also leads to lower RH. How would this influence the SOA formation?

**A1.4:** Thanks for reviewer's question. Higher temperature indeed led to lower RH due to the increased dew points. The literatures have suggested that although some studies have found that the variations of RH can influence the SOA formation, the influences were complex and in conflict. To reflect the question by the reviewer, we added the statement in the main text:

Line 598-606: "**In addition to the direct influences, the increase of temperature within OFR lead to the decreases of the relative humidity (RH), which can also impact SOA formation. However, the literatures show that the impact of RH on SOA formation remains inconclusive. For example, Tillmann et al. (2010) found the SOA yield was higher at humid conditions (RH: 40-70%) compared to dry conditions (RH: 0-10%) as the RH influenced the formation of products in α-pinene ozonolysis experiments. In contrast, Zhang et al. (2019) found the SOA yield of m-xylene-OH oxidation decreased as RH increased in a chamber study, as the high RH led to the less formation of oligomers and inhibited the reaction of $RO_2$ autoxidation. Thus, elucidating the influence of humidity on various SOA formations is still a challenge and falls outside the purview of our research topic here. In addition, given the short residence time within OFR (seconds to minutes), the impact of liquid phase reactions to SOA formation in OFR should be minimal.**"

**R1.5:** It is confusing when comparing Figure 3 and Figure 6b. (1) The horizontal distance is >400 mm in Fig. 3 but <200 mm in Fig. 6b. (2) The temperature shows a monotonic increasing trend from the inlet to the outlet in Fig. 3 but a minimum in the middle in Fig. 6b. Can the authors further explain the differences?

**A1.5:** To clarity, we added an illustration of vertical direction, horizontal direction, and probing direction (depth) in Fig. 1(b), as shown below. The horizontal and vertical directions formed a plane perpendicular to the probing direction (depth);

For the question (1), the x-axis in Fig. 3 was the probing depth from the inlet to outlet in the probing direction (460 mm in total, as shown in Fig. 1a). The x-axis of Fig. 6b shows the horizontal distance in horizontal direction (Fig. 6b). For the question (2), the x-axis of the two graphs did not indicate the same position. In Fig.3, all the temperatures were measured at the center line (the horizontal distance was at 0 mm) from the inlet to outlet in the probing direction. These positions were the same as the markers with a horizontal distance of 0 mm in Figure 6b, where a lower temperature at 100 mm were also shown.

[Figure]

**Figure 1: (a) Schematic plot for temperature measurement in the oxidation flow reactor of this study and (b) directions for temperature measurement. The center inlet, nut, and mesh screen near the front plate were removed when the temperature sensor was probed in the front direction. The information of different temperature sensors used can be found in Table S1.**

**R1.6:** Although PAM-OFR is the most commonly used OFR, there are many other types of OFRs. For other OFRs that put lamps outside of the reactor (like the ones listed

above), the heating issue is not as serious as PAM-OFR. Using the terminology "OFR" in the Conclusion may lead to misunderstanding. Therefore, I would suggest the authors use the terminology "PAM-OFR" rather than "OFR" throughout the manuscript.

**A1.6:** We replaced the "OFR" with "PAM-OFR" in the manuscript when the PAM-OFR was specifically referred to. This revision certainly makes our statements more rigorous.

---

## Author Comment (AC5)

**Response to reviewers for the paper "A comprehensive evaluation of enhanced temperature influence on gas and aerosol chemistry in the lamp-enclosed oxidation flow reactor (OFR) system" in Atmos. Meas. Tech. Discuss. Doi: 10.5194/amt-2023-230**

**By Tianle Pan et al.**

We appreciate the three reviewers' comments and support for the publication of this manuscript after revisions. Following the reviewers' suggestions, we have carefully revised the manuscript. To facilitate the review process, we have copied the reviewer's comments in black text. Our responses are in regular blue font. We have responded to all the referee comments and made alterations to our paper (**in bold text**).

**Anonymous Referee #3**

**General Comments**

Tianle Pan et al. present a study on the influence of the heating effect of oxidation flow reactor (OFR) UV lamps on oxidant chemistry, flow conditions and secondary organic aerosol (SOA) formation. The authors find that while the increase in temperature does not greatly affect the oxidant chemistry, the effect of temperature gradient on residence time distribution shortens the mean residence time. Thus, the OH exposure at higher light intensities is affected by the heating effect. The temperature increase in the OFR influences the SOA yields and composition, so that the SOA yield is lower at higher temperatures and the O to C ratio of SOA is higher at higher temperatures.

The heat caused by the OFR lamps is an important aspect of OFR studies that has not been well characterized earlier. The measurements and analysis in this study are comprehensive and thoroughly done. However, the quality of reporting and language is not fully adequate. Another concern is the relevance of the SOM model for the SOA formation in the OFR. Thus, I recommend publishing this study in Atmospheric

Measurement Techniques, but only after the following remarks have been addressed. Especially the language needs to be improved.

**Major Specific Comments**

**R3.1:** The reporting should be concise, and all statements should be well-defined and unambiguous. Currently, the text contains unnecessary extra words (like 'around', 'might'), which are sometimes necessary but could be mostly removed. The authors often use word 'support', which in many places could be replaced by a more exact wording. Furthermore, for many sentences in Section 3 it is unclear what the authors actually mean. Some of these are included in the following specific comments, but I recommend the authors read through the paper and for each sentence ponder whether the sentence is necessary, and whether the statement is unambiguous.

**A3.1:** We appreciated the reviewer's suggestions and comments. We carefully examined the manuscript sentence by sentence and revised the whole manuscript thoroughly. As shown in the tracked version of the manuscript, most of the sentences (>70%) were carefully revised. Here are some examples:

(1) Line 399-401: "**This indicated that the primary T/RH sensor in the backplate with default OFR settings lead to underestimation of the temperature inside of OFR, which should be verified and corrected by the users based on the configuration of their instrument.**" We replaced "**suggests**" with "**indicated that**" and "**might lead**" with "**lead**"

(2) Line 243-245: "**These results indicated that the temperature increase inside of the PAM-OFR was mainly due to the heat from the lamps, which was further confirmed by Fig. 3b**". We replaced "**supported**" with "**indicated**" and "**confirmed**".

(3) Line 255: "a**ssuming a mixing water ratio of 1.88%**" and Line 255: "**assuming ambient OH concentration of 1.5x106 molecules cm-3**". We deleted "**around**" before "**1.88%**" and "**1.5 x106**".

(4) Line 271-273: We modified "**This inconsistency is mainly due to the lamps starting at 10 V with colder conditions (e.g., room temperatures or lower**

**voltage settings), meanwhile, the OFR reactor has a thermal mass that needs time to accumulate or dissipate heat.**" to "**This discrepancy was primarily attributed to the fact that the lamps were initiated at 10 V under cooler conditions (e.g., room temperatures or lower voltage settings), while the OFR reactor had a thermal mass that required time to accumulate or dissipate heat.**"

(5) Line 629: "**cold air can also be effective ways**". We replaced "**might**" with "**can**".

**R3.2(a):** It is unclear why the authors used the ring flow manifold in RTD measurements when it was not used in the simulations. In my opinion, this is justified since the ring flow is typically used in actual measurements. However, the authors could discuss this justification when they describe the RTD measurement and simulation. This difference between measurement and simulation should also be mentioned in Fig. 8 caption.

**A3.2(a):** Thanks for reviewer's comments. Previously, we measured the RTD by sampling through ring flow. Then, we also realized that the simulation and measured results shall use the same flow set. When we submitted our manuscript to AMTD, the measured RTD results were already updated to these measured by sampling only from aerosol line at 5 L min⁻¹. However, we forgot to change the discussion. Thus, the modelled and measured RTD shown in Fig. 8 shall be consistent with each other. The sentence of "**This inconsistency is probably because i) we only considered the airflow sampled from the center outlet in the backplate, but not the ring flow manifold, which caused more recirculation**" was deleted.

**R3.2(b):** In addition, I think the description of RTD measurement (p. 13, l. 412-415) should be moved to Section 2.

**A3.2(b):** Yes, we moved the description for the RTD measurement to Sec. 2.

"**In addition to the temperature measurement in OFR, we measured the residence time distribution (RTD) at different voltages to explore the effect of temperature on RTD. Specifically, we first turned on the lights to make the temperature stable. Then, a 2 s pulse of 50 ppm SO₂ was injected into a 5 L min⁻¹**

**carrier gas (N₂) with RH <10%. N₂ was selected as the carrier gas to prevent the reaction between SO₂ and the generated oxidant when UV lamps were turned on. We measured the RTD with lamp driving voltage set at 0 V, 5 V and 10 V. Note that we only used the outlet for aerosol line for sampling (5 L min⁻¹) during the RTD measurement for better comparison with simulation results in Sec. 2.3."**

**R3.2(c)**: What were the flow rates in aerosol line and exhaust line during RTD or temperature measurements? The authors provide the flow rates in different occasions, but it is not clear whether this is the total flow rate (aerosol line + exhaust line) and what is the ratio between the aerosol line and exhaust line.

**A3.2(c):** Most of the time, sampling through the exhaust line was used for the temperature experiments. However, the ratio between aerosol line and exhaust line shall play very minor impact on the absolute temperature enhancement measured in side of OFR. We clarify the flow sampling strategy in the maintext: "**Most of the temperature experiments were done with sampling exhaust line from the ring flow.**". For the RTD experiment, the flow sampled from the aerosol lines was displayed in Fig. 8. We clarify this in the main text "**Note that we only used the outlet for aerosol line for sampling (5 L min⁻¹⁾ during the RTD measurement for better comparison with simulation results in Sec. 2.3.**"

**R3.3:** The authors use term 'Enhanced temp.' in figures to describe the difference between OFR and ambient temperature. In my opinion, e.g. 'Temp. enhancement' or just $\Delta T$ would be better.

**A3.3** We replaced all the 'Enhanced temp.' with $\Delta T_{(OFR-amb.)}$ in the figures in the revised manuscript.

**R3.4:** When discussing the approaches to reduce the heating effect in Sect. 3.5, the authors actually only discuss approaches to reduce the effect of increased temperature. For example, when the fans are used to cool down the OFR external surface, the heat transfer inside the OFR is improved and this affects the RTD since the heat transfer occurs via convection. However, the authors did not characterize the effect of different

cooling methods on the RTD, which at least should be mentioned in this section or the section headline should be changed.

**A3.4:** We appreciated the reviewer's reminding. This is a really good point. Corresponding explanation about the potential influence on RTD was added in this section. **"Cooling down the OFR would also affect RTD since the heater transfer occurs via convection inside of OFR, which needs to be further investigated in the future."**

**R3.5:** The authors sometimes mix past and present tense. See e.g. Fig. 3 caption: four lamps were turned on, flow rate is 5 lpm.

**A3.5:** We examined all the tenses used in the text and made corresponding corrections. All the revisions can be seen in the tracked version of manuscript.

**R3.6:** Based on the current description of the SOM model, it seems that the authors first model the SOA formation in "normal" temperature and then study its evaporation in elevated temperature (e.g. p. 17, l. 494: "for the newly formed SOA in the OFR, the temperature impact was simulated based on SOM model"). In that case, what is the temperature where the SOA formation is modeled? If this is the case, the model results regarding e.g. the SOA yield are not very relevant, because in the OFR the SOA formation would take place in the elevated temperature.

The authors should change the heading of Section 3.4 to "Temperature influence on OA evaporation", and describe more clearly that they are modeling the evaporation of SOA that was formed in temperature X and then injected into the OFR at elevated temperature. This needs rewriting of Sect. 3.4 so that the authors discuss OA evaporation instead of SOA formation.

**A3.6:** I think there is a misunderstanding. The OA evaporation and SOA formation are two separate topics in our discussions. We simulate the SOA formation using SOM under different temperatures directly. No OA seed was considered. To clarify, we separate the original section 3.4 into two sections, which is "**3.4 Temperature influence on the evaporation of ambient OA**" and "**3.5 Temperature influence on**

**the SOA formation**".

Corresponding explanations were added in the section 3.4: "**Here, we estimated the potential losses of input ambient OA due to evaporation under enhanced temperature in OFR. This estimation is mainly based on literature results and modeling work.**"

Specific comments

**R3.7**: p. 1 l. 44: box model using radical chemistry → radical chemistry box model
**A3.7:** Corrected

**R3.8**: p. 2 l. 82: The high temperature inside the OFR does not cause the recirculating flows, it is the temperature gradient caused by lamp heating.
**A3.8:** Corrected

**R3.9**: p. 3 l. 88: SOA simulation and study → SOA simulations and studies
**A3.9:** Corrected

**R3.10**: p.3 l. 102: The acronym ARI has not been declared earlier.
**A3.10:** We added the definition in line 101: "**The PAM-OFR (Aerodyne Research, Inc., abbreviated as ARI) used in this study…**"

**R3.11:** p. 4, Fig. 1: What is the exhaust gas line? It seems that sensor (3) is measuring the ring flow outlet, but in the text it is unclear whether the exhaust gas means the ring flow or the N2 purge flow. Please clarify.
**A3.11:** The exhaust gas line means the ring flow. We added the explanation in line 336 and specify the exhaust gas line come out from ring flow in Fig.1:

Line 338: "The dissipation of energy through the exhaust air **(from the ring flow)** from the PAM-OFR"

[Figure]

**Figure 1: (a) Schematic plot for temperature measurement in the oxidation flow reactor of this study and (b) directions for temperature measurement. The center inlet, nut, and mesh screen near the front plate were removed when the temperature sensor was probed in the front direction. The information of different temperature sensors used can be found in Table S1.**

**R3.12:** p. 7 Fig. 2 caption: "assuming ambient OH concentrations are around 1.5e6..." I suppose you have used an exact value of 1.5e6 in the calculations, so please remove the word 'around' (also earlier regarding the water mixing ratio).

**A3.12:** We removed the 'around' in the caption of Fig.2. And we modified the expression in line 238 for mixing ratio: "**(assuming the water mixing ratio is 1.88%, RH=60%, external OH reactivity=30 s$^{-1}$).**"

**R3.13:** p. 10 l. 296: "In our case, the..." → "In our case, when the..."
**A3.13:** Corrected

**R3.14**: p. 10 l. 298: What is "vertical axial direction"? I think axial would mean the direction of the central axis of rotation. It would be helpful to define the different directional terms (vertical, horizontal, probing depth) graphically e.g. by adding another panel in Fig. 1.

**A3.14:** Following reviewer's suggestions, we defined the direction of measurement in Fig. 1b and modified the text in line 126-128: "**Briefly, we measured the air temperature inside PAM-OFR at varied positions (vertical and horizontal**

**directions, as well as different depths from inlet (Fig. 1b)) under different lamp configurations (e.g., number of lamps, types of lamps, intensity of lamps) and flow rates.**"

[Figure]

**Figure 1: (a) Schematic plot for temperature measurement in the oxidation flow reactor of this study and (b) directions for temperature measurement. The center inlet, nut, and mesh screen near the front plate were removed when the temperature sensor was probed in the front direction. The information of different temperature sensors used can be found in Table S1.**

**R3.15**: p. 11 l. 319: "OFR chambers (OFR metal tubes)" → "OFR surface"

**A3.15:** We replaced the "OFR chambers (OFR metal tubes)" with "OFR surface".

**R3.16**: p. 11 l. 331: surfaces → surface

**A3.16:** Corrected.

**R3.17**: p. 11 l. 333-339: The authors discuss in length why the temperature of the external surface is lower than that of the gas inside the reactor, while it is basic physics that since the OFR is not isolated and the ambient temperature is lower than the OFR internal temperature, there will be heat transfer from inside the OFR to the ambient, and the medium between these (the surface) will be at lower temperature than the OFR internals. This is correctly described in lines 338-339. The thermal mass of the OFR is not relevant, as it affects only the rate of temperature increase but not the final equilibrium temperature that is discussed here.

**A3.17:** We agree with the reviewer's comments on balanced condition. However, when the temperature equilibrium between OFR tube and air was not balanced, the thermal mass would be a reason as well. To reflect that we move the original cause "ii" (heating transfer) to be cause "i". Then explain the original cause "i" only work when the thermal system was not balanced.

**" ii) When the temperature equilibrium between the air and OFR was not balanced, an additional reason will cause the lower temperature in OFR tube. The aluminum OFR chamber has a higher thermal mass than the air. Although the specific heat capacity of metal (0.908 J $g^{-1}$ $K^{-1}$ at 301.60 K) is similar to that of air (1.005 J $g^{-1}$ $K^{-1}$ at 300 K) (Giauque and Meads, 2002; Kieffer, 1956), the flow tube is considerably heavier than the air due to its significantly higher density (2700 Kg $m^{-3}$ vs 1.29 Kg $m^{-3}$), resulting in a lower temperature for the OFR tube than the inner air."**

---

## Author Response (AR2)

**Response to reviewers for the paper "A comprehensive evaluation of enhanced temperature influence on gas and aerosol chemistry in the lamp-enclosed oxidation flow reactor (OFR) system" in Atmos. Meas. Tech. Discuss. Doi: 10.5194/amt-2023-230**

**By Tianle Pan et al.**

We appreciate the three reviewers' comments and support for the publication of this manuscript after revisions. Following the reviewers' suggestions, we have carefully revised the manuscript. To facilitate the review process, we have copied the reviewer's comments in black text. Our responses are in regular blue font. We have responded to all the referee comments and made alterations to our paper (**in bold text**).

**Anonymous Referee #3**

The authors have significantly improved the language and satisfactorily responded to some of my comments. However, I cannot find this revision acceptable because the authors did not respond to all of my remarks. I submitted 4 pages of comments, but the authors only responded to the comments in the 2 first pages.

We are sincerely sorry that we forgot the last two pages in the last round of revision. Here we attached the rest responses from last round. The sequence number for each comment was continued with the first two pages and start with question "R3.18". The responses for the new comments are also addressed point by point in the following.

**R3.18**: p. 12 l. 352: Here and elsewhere the authors use term 'OFR tubes' for the OFR casing. I would suggest finding a better term, since to my understanding this structure consists of a single large tube and the end plates, not 'tubes'.

**A3.18:** We replaced all the "OFR tubes" with "**OFR enclosure**".

**R3.19**: p. 13 l. 395: was → is

**A3.19:** Corrected.

**R3.20**: p. 13 l. 396: What is 'more even and better-mixed air'? I think the purpose of the ring flow manifold is to reduce dead space at the end of the OFR and thus reduce the mixing.

**A3.20:** Yes, the ring flow manifold can reduce the dead space within the OFR. In our text, "more even and better-mixed air" means the airflow within OFR is more uniform as there is less recirculation, which also leads to a better mixing for the gas within OFR, such as the mixing between the precursors and oxidant. In order to reduce ambiguity, we have made modifications in the text:

**"In addition to aerosol sampling, the air for gas phase measurements is usually sampled through a perforated ring flow manifold in the back of the PAM-OFR to reduce wall effects and recirculation, which makes the airflow more stable and uniform (Fig. S1)."**

**R3.21**: p. 13 l. 397: Please check the tense of this sentence. Also, the flow distribution certainly will be different, so the word 'might' is not necessary here.

**A3.21:** Corrected.

**R3.22**: p. 13 l. 409-411: The definition of "sum of integrals of signal over time versus the sum of signals" is not correct. Please define the mean residence time correctly or just cite the relevant source.

**A3.22:** We deleted the description of the average residence time calculation.

**"Details of the $\tau_{avg}$ calculation can be found in Huang et al. (2017)."**

**R3.23**: p. 13 l. 412: Here and elsewhere: the authors have measured RTDs and thus also the mean residence times. "Average RTD values" is not correct term.

**A3.23:** We replaced the "Average RTD values" with **"average residence time ($\tau_{avg}$)".**

**R3.24**: p. 14 l. 427: calculation $\rightarrow$ simulation?

**A3.24:** Changed to "**simulation**".

**R3.25(a)**: p. 14 Fig. 7: In panels a, c and e, there are 4 different cases in each panel. I

assume these represent the situation at probing depths of 100, 200, 300 and 400 mm. Please indicate this somehow in the figure or in the caption.

**A3.25(a):** Thanks for the suggestion, we added the statement in both the figure and caption.

[Figure]

**Figure 7: (a) Three-dimensional simulation results demonstrating the cross-sectional temperature profiles. Four cross-sectional figures from left to right represented the results at probing depths of 100, 200, 300, and 400 mm for panel (a), (c), and (e). (b) Simulated lateral temperature profile inside of the OFR based on CFD simulation. Panel (a) and (b) were simulated with conditions that four**

**lamps were set to be 5 V. (c) Three-dimensional simulation results illustrating cross-sectional velocity profiles and (d) lateral velocity profiles based on CFD simulation results. Panel (c) and (d) were simulated with conditions that lamps were off (room temperature: 25 ℃). (e-f) the same plots as panels (c-d) with four lamps set to 5 V. One-dimensional axial velocity profiles at (g) horizontal and (h) vertical directions inside the OFR at room temperature, and one-dimensional axial velocity profiles at (i) horizontal and (j) vertical directions with four lamps set to 5 V. The positive values indicated the velocity direction from inlet to outlet (Forward) and the negative values represented the velocity direction from outlet to inlet (Backward).**

**R3.25(b)**: In panels g-j: What does the horizontal or vertical velocity mean? If these are the velocities towards the horizontal or vertical direction, then the labels 'backward' and 'forward' do not make sense. Or should it be axial velocity?

**A3.25(b):** They are the axial velocity at horizontal or vertical directions. The positive values of velocity indicate the velocity direction from inlet to outlet ('Forward' in the graph). We added the explanation in the caption as shown in A3.25(a).

**R3.25(c)**: Figure caption: "Enhanced temperature influences in panels (c) and (d) were considered" – should this be panels (e) and (f)? The last reference to panel (h) should be (j).

**A3.25(c):** Yes, we meant that panels (e) and (f) were the results with enhanced temperature influences based on panels (c) and (d). We deleted this sentence for clarity. We also revised the caption for the panel (j). Results could be seen in A3.25(a).

**R3.26**: p. 15 Fig. 8: The y axis label should be "Probability density function" (or Residence time distribution) with unit of $s^{-1}$, not "probability distribution function". In figure caption, "The average RTD values were also estimated here" → "The mean (average) residence times are also shown here". The calculation of the mean residence time is not an estimate, but a well-defined property of the RTD.

**A3.26:** We revised the y axis label to "**Probability density function (s⁻¹)**" and modified the caption as "**The average residence times are also shown here**".

[Figure]

**Figure 8: Residence time distribution (RTD) of SO₂ within the PAM-OFR under different lamp settings. A 2 s pulser of SO₂ was injected into the OFR. The average residence times are also shown here. The simulated results from CFD model are shown in red lines.**

**R3.27**: p. 15 l. 456: RTD distributions → RTDs

**A3.27:** Corrected.

**R3.28**: p. 15 l. 460: reaction rate**s**, concentration**s** of

**A3.28:** Corrected.

**R3.29**: p. 16 l. 467: "very minor to negligible impact on the gas-phase reactions". This is too strong a statement since only a limited set of gases and their reaction pathways was studied.

**A3.29:** We modified the sentence: "**Our results suggested that an increase in temperature within OFR due to lamp heating would have a minimal impact on gas-phase reactions**".

**R3.30**: p. 16 l. 485: per (°C second) $\rightarrow$ s$^{-1}$°C$^{-1}$

**A3.30:** Corrected.

**R3.31**: p. 16 l. 487: mass loss of ~32% for ambient OA

**A3.31:** We revised the sentence.

**R3.32**: p. 17 l. 492: "might" is unnecessary here

**A3.32:** Corrected.

**R3.33**: p. 17 l. 519-520: unclear sentence

**A3.33:** We modified the sentence:

**"Note that in the model we did not specifically treat the temperature effect on autoxidation reaction rate of RO$_2$. E.g., varied yields of highly oxygenated organic molecules (HOMs) from autoxidation shall be used at different temperatures, while constant yields for HOMs were used during the simulation in this study."**

**R3.34**: p. 17 l. 521: "Using a constant yield for HOMs..." $\rightarrow$ "The constant yield for HOMs used in this model might..."

**A3.34:** We revised the sentence according to the reviewer's suggestion.

**R3.35**: p. 18 l. 534: "As the detailed..." $\rightarrow$ "The detailed..."

**A3.35:** Corrected.

**R3.36**: p. 19 l. 557-558: Unclear sentence. Consider e.g. "Increasing Hvap increases the sensitivity of SOA formation to temperature".

**A3.36:** We revised the sentence according to the reviewer's suggestion.

**R3.37**: p. 19 l. 561: what does "based on different species and temperature" mean?

**A3.37:** We revised the sentence: **"These variations were observed across different precursors and temperatures."**

**R3.38**: p. 19 Fig. 11: Is the 30 µg/m3 the inlet concentration? And is the x-axis in panels

b and d the inlet concentration of OA?

**A3.38:** Yes, both concentrations are the inlet concentration. We modified the captions of Fig.11 and other graphs to point out the OA mass concentration is inlet concentration.

[Figure]

**Figure 11: Size distribution of dodecane SOA at different temperatures under (a) high NOx and (c) low NOx conditions, respectively. A mass concentration of 30 μg m⁻³ for OA seed (inlet mass concentration) was assumed for size distribution simulation here. The O:C ratio of dodecane SOA as a function of temperature and mass concentration of OA seed (inlet mass concentration) under (b) high and (d) low NOx conditions. The equivalent aging time was 1 day by assuming the ambient OH concentration equated to 1.5×10⁶ molecule cm⁻³ (Mao et al., 2009).**

**R3.39**: p. 19 Fig. 12: "the ambient OH concentration"
**A3.39:** Corrected.

**R3.40**: p. 20 l. 572-573: "which shows varied extents for different precursors and reaction conditions" – what does this mean?
**A3.40:** We modified this sentence as:

**"In summary, the heating effect induced by the lamps could significantly influence the SOA formation within OFR for certain high-OH exposure applications. This impact of temperature varied depending on the specific precursors and reaction conditions."**

**R3.41**: p. 20 l. 573: deceased → decreased

**A3.41:** Corrected.

**R3.42**: p. 20 l. 583: reducing → reduce

**A3.42:** Corrected.

**R3.43**: p. 20 l. 590: deduction → reduction

**A3.43:** Corrected.

**R3.44**: p. 20 Fig. 13: bellowing → blowing

**A3.44:** Corrected.

**R3.45**: p. 22 l. 621: trends → hystereses?

**A3.45:** We replaced "trends" with "**concentration curves**".

**R3.46**: p. 22 l. 639-640: is the shorter residence time really due to accelerated diffusion?

**A3.46:** The temperature gradients in the OFR caused by the UV lamps lead to the axial dispersion (the strengthening effect of molecular diffusion) and thus influence the residence time (Lambe et al., 2019). We revised the text:

**"The pulsed tracer measurements suggested that the increased temperature in the PAM-OFR induced the axial dispersion (Lambe et al., 2019), leading to a shorter average residence time ($\tau_{avg}$)"**

**Anonymous Referee #3 (New comments for the second rounds)**

**R3.47**: In addition, regarding my earlier comment (R3.6) and the authors' response (A3.6), there are still conflicting statements in the manuscript. In their response, the

authors say that the SOA formation was simulated under different temperatures directly, but the text in the manuscript does not support this. Some examples are listed below.

**A3.47:** Thank you for your comments. We revised point-by-point based on reviewer's suggestion.

**R3.47**(a): -The authors say in their response that no OA seed was considered. However, in p. 7, l. 221 and p. 18, l. 519 they write that the SOA formation was modeled under different OA concentrations (1-80 µg/m3).

**A3.47(a):** Sorry for the mistake in R3.6, we meant that we did not consider the evaporation and reaction for the OA seed in simulation. We have revised the description in the main text:

Line 220-222**: "In this study, SOA formation from four typical VOC precursors including dodecane, α-pinene, toluene, and m-xylene was modeled under different OA seed concentrations (1-80 µg m$^{-3}$) and NOx conditions (low NOx vs high NOx). We did not consider the evaporation and reaction of OA seed in the model."**

Line 520-521: **"The evaporation and chemical reaction of existing OA seed under different temperatures were not considered in the model."**

**R3.47**(b): -p. 18, l. 516: "For the newly formed SOA in the OFR, the temperature impact was simulated..." So here SOA is first formed (independent of temperature), and then the temperature effects on that SOA are studied? I think the temperature effect on the SOA formation itself should be studied.

**A3.47(b):** Actually, we simulate the SOA formation under different temperatures directly. We revised the sentence:

Line 517-518**: "The formation of SOA from the oxidation of different VOC precursors within OFR was simulated at different temperatures using the SOM model".**

**R3.47**(c): -p. 18, l. 524: "the higher temperatures result in lower SOA yields due to the increased evaporation of gas-phase products." I disagree. I think at higher temperatures

some of the oxidation products never condense onto particle phase, which leads to a lower SOA yield. So some compounds just stay in the gas-phase, which does not mean evaporation. Again, here it seems like a constant SOA mass is formed independent of temperature, and then the evaporation is modeled at different temperatures.

**A3.47(c):** We agree with the reviewer. The sentence needs to be clarified. We modified the sentence:

Line 526-527**: "the higher temperatures result in lower SOA yields due to the increased partitioning of oxidation products in gas-phase".**

**R3.47**(d): -p. 20, l. 576: Here the authors address the evaporation as well.

**A3.47(d):** We revised the sentence:

Line 579-581**: "The higher O:C at higher temperatures was probably caused by the less partitioning of semi-volatile and less-oxidized components into particle phase with increased temperatures".**

**R3.47**(e): -p. 20, l. 579: Also here the authors talk about evaporated SOA mass.

**A3.47(e):** We revised the sentence:

Line 582-583**: "In the model, the gas-particle partitioning of oxidation products as a function of temperature was mainly determined by the enthalpy".**

**R3.48**: -p. 18, l. 514: "Obvious SOA decrease mass was observed in OFR at different temperatures." Please rephrase this, or in my opinion this sentence is not needed here, since the authors later report the results of the laboratory experiments (p. 18, l. 529).

**A3.48:** We deleted the sentence as the reviewer's suggestion.

**Reference**

Huang, Y., Coggon, M. M., Zhao, R., Lignell, H., Bauer, M. U., Flagan, R. C., and Seinfeld, J. H.: The Caltech Photooxidation Flow Tube reactor: design, fluid dynamics and characterization, Atmospheric Measurement Techniques, 10, 839-867, https://doi.org/10.5194/amt-10-839-2017, 2017.

Lambe, A. T., Krechmer, J. E., Peng, Z., Casar, J. R., Carrasquillo, A. J., Raff, J. D., Jimenez, J. L., and Worsnop, D. R.: HOx and NOx production in oxidation flow reactors via photolysis of isopropyl nitrite, isopropyl nitrite-d7, and 1,3-propyl dinitrite at $\lambda$ = 254, 350, and 369nm, Atmospheric Measurement Techniques, 12, 299-311, https://doi.org/10.5194/amt-12-299-2019, 2019.

Mao, J., Ren, X., Brune, W. H., Olson, J. R., Crawford, J. H., Fried, A., Huey, L. G., Cohen, R. C., Heikes, B., Singh, H. B., Blake, D. R., Sachse, G. W., Diskin, G. S., Hall, S. R., and Shetter, R. E.: Airborne measurement of OH reactivity during INTEX-B, Atmospheric Chemistry and Physics, 9, 163-173, https://doi.org/10.5194/acp-9-163-2009, 2009.